# Adapting While Learning:
# Grounding LLMs for Scientific Problems with Tool Usage Adaptation

**Bohan Lyu** [* 1]   **Yadi Cao** [* 2]   **Duncan Watson-Parris** [2]   **Leon Bergen** [2]   **Taylor Berg-Kirkpatrick** [2]   **Rose Yu** [2]

## Abstract

Large Language Models (LLMs) demonstrate promising capabilities in solving scientific problems but often suffer from the issue of hallucination. While integrating LLMs with tools can mitigate this issue, models finetuned on tool usage become overreliant on them and incur unnecessary costs. Inspired by how human experts assess problem complexity before selecting solutions, we propose a novel two-component fine-tuning method, **Adapting While Learning** (**AWL**). In the first component *World Knowledge Learning* (WKL), LLMs internalize scientific knowledge by learning from tool-generated solutions. In the second component *Tool Usage Adaptation* (TUA), we categorize problems as easy or hard based on the model's accuracy, and train it to maintain direct reasoning for easy problems while switching to tools for hard ones. We validate our method on 6 scientific benchmark datasets across climate science, epidemiology, physics, and other domains. Compared to the original instruct model (8B), models post-trained with AWL achieve 29.11% higher answer accuracy and 12.72% better tool usage accuracy, even surpassing state-of-the-art models including GPT-4o and Claude-3.5 on 4 custom-created datasets. Our code is open-source at https://github.com/Rose-STL-Lab/Adapting-While-Learning.

## 1. Introduction

To realize the ultimate dream of building an AI scientist, numerous works have explored the impressive capabilities of large language models (LLMs) in solving scientific problems, from answering general questions (Lu et al., 2022; Zhang et al., 2024b) to contributing to scientific discoveries (Ma et al., 2024; Kumar et al., 2023; Liu et al., 2022). However, except for the largest models like ChatGPT-o1 and DeepSeek-v3, the abilities of LLMs for scientific reasoning are still typically limited to high school levels (Rein et al., 2024; Cobbe et al., 2021; Hendrycks et al., 2024). LLMs have the innate behavior of hallucination (Farquhar et al., 2024) and can produce scientifically invalid outputs.

Recent studies have shown that LLMs can mitigate hallucination by accessing general-purpose tools (Schick et al., 2023; Tang et al., 2023; Patil et al., 2023; Qin et al., 2023; Wang et al., 2024b). Naturally, incorporating specialized scientific tools, such as physics-based simulators, presents a solution to complex scientific problems (Schick et al., 2023; Ma et al., 2024; Liu et al., 2022). However, recent studies also indicate that LLMs lack the ability to make adaptive decisions about tool use (Yu et al., 2024; Huang et al., 2023): for hard problems, LLMs may not know when or how to use tools, resulting in hallucinatory responses; conversely, for easy problems, LLMs may become over-reliant on tools, resulting in unnecessary computational cost overheads.

We observe that human scientists often first gauge the difficulty of a problem before deciding whether to direct reason or employ external tools (Payne et al., 1993; Kruger & Dunning, 1999). Hence, we seek to instill similar adaptive capabilities in LLMs to achieve a balance between accuracy and cost when solving scientific problems.

To this end, we propose a novel training paradigm, Adapting While Learning (AWL), which consists of two components. The first component, **World Knowledge Learning (WKL)**, uses supervised fine-tuning and preference learning to align a pre-trained LLM with highly accurate solutions generated using information from external tools, thereby internalizing scientific knowledge. In the second component, **Tool Usage Adaptation (TUA)**, we evaluate the LLM's direct answering ability and classify questions as easy or hard based on the model's accuracy. While maintaining the same alignment target for easy questions, we train the model to follow external tool-usage traces for hard questions, enabling intelligent selection based on problem complexity.

---

[*]Equal contribution  [1]Tsinghua University. Work partially done during Bohan Lyu's visit to UC San Diego.  [2]University of California, San Diego. Correspondence to: Rose Yu <roseyu@ucsd.edu>.

*Proceedings of the $42^{nd}$ International Conference on Machine Learning*, Vancouver, Canada. PMLR 267, 2025. Copyright 2025 by the author(s).

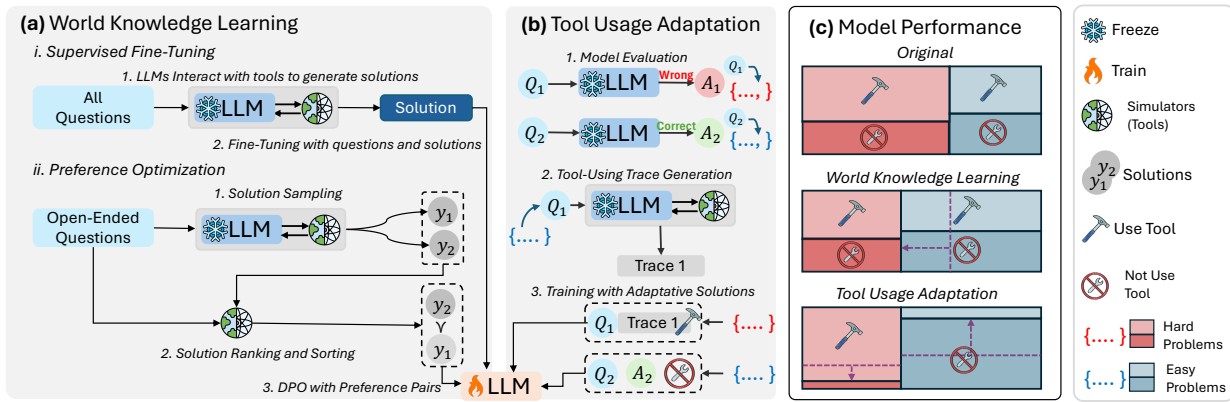

*Figure 1.* **Pipeline of Adapting While Learning.** (a) World Knowledge Learning: LLMs undergo supervised fine-tuning for all questions and preference learning for open-ended questions; (b) Tool Usage Adaptation: Questions are classified as easy/hard based on LLM's direct-answer accuracy. For easy questions, training targets remain unchanged as the solutions; for hard questions, targets are modified to tool-usage traces. (c) Model improvement visualization: Leftward movement of the vertical dashed line indicates more questions can be solved internally; Movements of horizontal lines for easy/hard questions, respectively, show more intelligent tool usage decisions.

We empirically evaluated our model on a diverse range of scientific datasets, from college-level math and physics to research frontiers like climate science and epidemiology. The experimental results show significant improvements for a pre-trained base model (8B) after post-training with our method. The post-trained model even surpasses frontier closed models on our newly created custom datasets containing challenging and specialized questions that frontier LLMs had not encountered during their pre-training.

Our contributions are summarized as follows:

- We introduce a novel two-component training paradigm, Adapting While Learning (AWL), which enables LLMs to efficiently solve real-world scientific problems of varying complexity.

- We construct 4 new datasets spanning various scientific domains: epidemiology, climate, Mojuco, and PDEs, to facilitate future research in this direction.

- Compared to the pretrained base model, our post-training achieves an average improvement of 29.11% in answer accuracy and a 12.72% increase in tool usage accuracy across all datasets. On our newly created datasets, the post-trained model even surpasses state-of-the-art closed models like GPT-4o and Claude-3.5.

## 2. Related Work

**LLM Alignment.** Alignment techniques aim to make LLMs behave in accordance with human values, using methods such as supervised fine-tuning (SFT) (Zhang et al., 2024a; Scheurer et al., 2023; Dong et al., 2023; Yuan et al., 2023; Song et al., 2024) and reinforcement learning

(RL) (Rafailov et al., 2024; Meng et al., 2024; Ouyang et al., 2022; Lee et al., 2023; Bai et al., 2022). Direct Preference Optimization (DPO) (Rafailov et al., 2024) is a special replacement to RL that utilizes designed preference between pairwise data for alignment, which makes it particularly suitable for data collection for post-training.

In our work, we employ SFT for all questions and additionally utilize DPO to learn preferences between different proposals for open-ended questions.

**Training LLMs for Scientific Problems.** Previous work has sought to ground LLMs using domain-specific knowledge in various scientific fields: climate science (Thulke et al., 2024), biomedical science (Luo et al., 2022), molecular science (Chithrananda et al., 2020), and general science (Zhang et al., 2024b; Taylor et al., 2022). Most of these approaches heavily rely on expert annotations or distillation from stronger models and face scalability limitations due to computational and expert labor costs.

These limitations highlight the need to integrate scientific tools into both data generation and training processes.

**LLM Tool Usage.** LLMs have demonstrated impressive performance in using external tools (Yao et al., 2023; Schick et al., 2023; Patil et al., 2023; Qin et al., 2023), such as web interfaces and shopping platforms (Yao et al., 2022; Cheng et al., 2024), code interpreters (Ma et al., 2024; Cai et al., 2023), scientific simulators (Kumar et al., 2023; Liu et al., 2022; Bran et al., 2023), and scientific knowledge bases (Kraus et al., 2023; Thulke et al., 2024).

Among these tools, scientific tools (simulators and knowledge bases) provide consistent results that could potentially be internalized by the model, yet prior works have not ex-

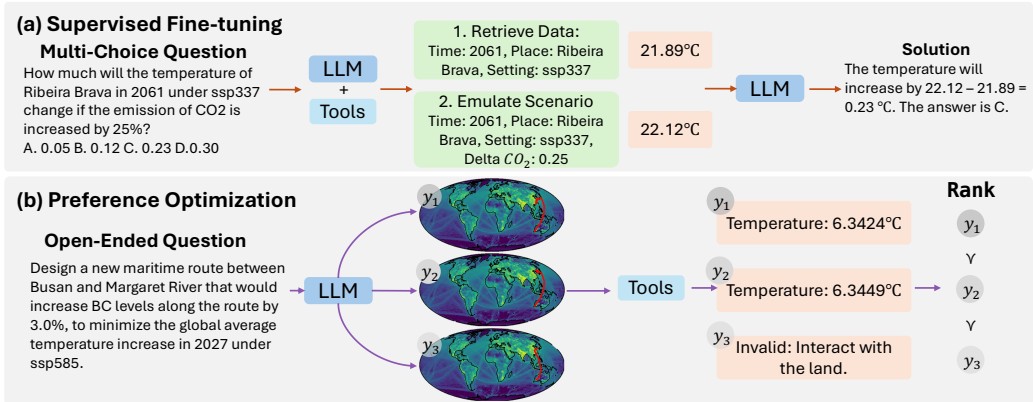

*Figure 2.* WKL training strategies: (a) For both determinate and open-ended questions, we first train the LLM to directly generate solutions (obtained from tool interactions during the solution generation phase). (b) For open-ended questions, additionally, we sample an ensemble of trial proposals, rank the proposals using predefined metrics, and convert the rankings into preference pairs for DPO training.

plored this opportunity. Furthermore, existing studies have not addressed training LLMs to make adaptive decisions about tool usage based on problem complexity, often resulting in over-reliance on the tools covered during training.

These limitations highlight the need for a training approach that enables LLMs to use tools adaptively and reach a balance between answering with internal knowledge and seeking help from external tools.

# 3. Methodology

As shown in Figure 1, our pipeline first generates solutions through tool interactions for each question (Section 3.1). The training process consists of two components: World Knowledge Learning (WKL), where the model is trained to internalize the knowledge directly (Section 3.2), and Tool Usage Adaptation (TUA), which classifies questions as easy or difficult based on the precision of the model's direct response without tools. We maintain direct-answer targets for easy questions, while changing the training targets to tool traces for hard questions (Section 3.3). To ensure the consistency of knowledge between components, we design a combined loss across WKL and TUA (Section 3.4). Finally, we extend the framework to open-ended questions by incorporating preference optimization (Section 3.5).

## 3.1. Generating Solutions and Tool Traces

As shown in Figure 2, we developed an automated solution generation pipeline that produces both direct responses and tool-use traces. The LLM $\pi$ receives access to scientific tools $E$ (e.g. numerical simulators) via system prompts. Given the context of the question $x$ with a labeled, correct tool trace $t$, we force the LLM to use the tools through $t$ by prompt $P_f$. The LLM then generates a solution $y$ by combining the context $x$ with the returned information from

trace $t$: $\{I_E\}_t$. Both the solution $y$ and the tool trace $t$ are the labels in our dataset, which can serve as training targets, respectively, depending on the difficulty of the question. The process can be formalized as:

$$y \sim \pi(\cdot \mid x, \{I_E\}_t, P_f). \tag{1}$$

## 3.2. World Knowledge Learning (WKL)

In WKL, we finetune a pre-trained model $\pi_\theta$, where $\theta$ represents the trainable parameters for finetuning, to generate solutions $y$ directly without tool usage. The no-tool-use restriction is specified in the prompt $P_n$. This process is formalized as:

$$\begin{aligned} J_{\text{Direct}}(\theta, \mathcal{D}, P) = \\ - \mathbb{E}_{x \sim \mathcal{D}, y \sim \pi(\cdot \mid x, \{I_E\}_t, P_f)} \left[ \log \pi_\theta(y \mid x, P) \right], \end{aligned} \tag{2}$$

where $\mathcal{D}$ represents the training dataset. The loss for WKL is then:

$$J_{\text{WKL}}(\theta, \mathcal{D}) = J_{\text{Direct}}(\theta, \mathcal{D}, P_n). \tag{3}$$

While WKL aims to internalize knowledge for direct problem solving, certain complex problems are still too challenging to learn. Therefore, we follow with TUA to train the model to intelligently switch to tools for these problems.

## 3.3. Tool Usage Adaptation (TUA)

TUA begins with partitioning the dataset into easy/hard subsets by evaluating the model after it has been trained with WKL for an epoch. For each question, we sample an ensemble of directly generated answers to calculate the accuracy rate. If the accuracy is higher than a predefined threshold, the question is classified as easy, resulting in two subsets: $\mathcal{D}_{\text{easy}}$ for problems the LLM can solve directly, and $\mathcal{D}_{\text{hard}}$ for the remaining ones. We now relax the constraint

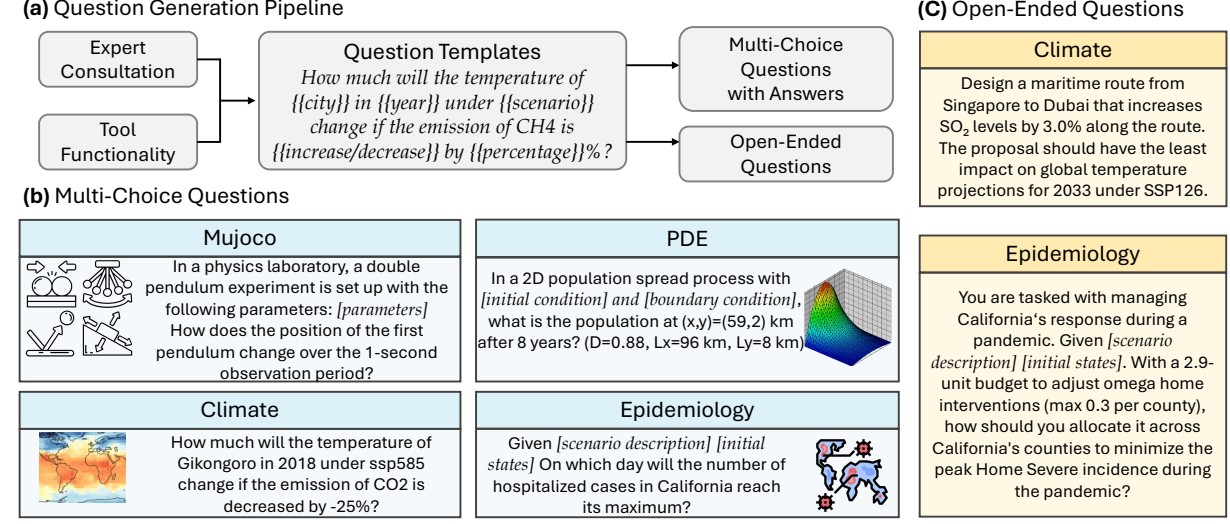

*Figure 3.* (a) Question generation pipeline using templates. Selected demo (b) multi-choice and (c) open-ended questions from our custom-created datasets.

on tool usage and let the model choose whether to use tools based on the question's difficulty (with prompt denoted as $P_i$). The success of making the correct binary decision is achieved implicitly by maintaining different alignment targets for easy/hard subsets: For easy problems $\mathcal{D}_{\text{easy}}$, we keep the alignment target as direct answering; however, for hard problems $\mathcal{D}_{\text{hard}}$, we switch the alignment target to the augmented solution with tool usage trace and guide the LLM to follow the trace $t$ outside of the tool set $E$. For $\mathcal{D}_{\text{hard}}$, the alignment loss reads:

$$
\begin{aligned}
J_{\text{Trace}}(\theta, \mathcal{D}, P) = \\
- \mathbb{E}_{x \sim \mathcal{D}, t \sim \pi(\cdot | x, \text{E}, P_f)} \log \pi_\theta(t \mid x, \text{E}, P).
\end{aligned}
\tag{4}
$$

The combined training loss considering both easy and hard questions in the whole dataset reads:

$$
\begin{aligned}
J_{\text{TUA}}(\theta, \mathcal{D}_{\text{easy}}, \mathcal{D}_{\text{hard}}) = \\
J_{\text{Direct}}(\theta, \mathcal{D}_{\text{easy}}, P_i) + J_{\text{Trace}}(\theta, \mathcal{D}_{\text{hard}}, P_i).
\end{aligned}
\tag{5}
$$

We note again that we apply the same prompt $P_i$ for all questions during TUA. This consistency is important for deployment, when we do not have labels for new questions, requiring the LLM to implicitly decide on its own whether to use tools or not (i.e., we must use $P_i$).

### 3.4. Knowledge Consistency Across Components

A naive approach to combining the two components would be to alternate the training loss terms between adjacent epochs. However, in our preliminary experiments, we observed significant performance drops for easy problems using that approach.

We attribute this performance drop to the loss of consistency between different prompts: As noted above, during deployment, only the prompt $P_i$ will be used to enable the adaptive switch to tools for more difficult problems; however, the internalization of knowledge for easy questions was only optimized using the prompt $P_n$. Recent work (Zeng et al., 2024) indicates that knowledge acquired under one prompt may not transfer well to another, potentially leading to performance degradation.

To address this problem, we propose a combined loss term that simultaneously optimizes the alignment for both components, eliminating the need for alternation between components:

$$
\begin{aligned}
J_{\text{Mix}}(\theta, \mathcal{D}, \mathcal{D}_{\text{easy}}, \mathcal{D}_{\text{hard}}) = \\
J_{\text{WKL}}(\theta, \mathcal{D}) + J_{\text{TUA}}(\theta, \mathcal{D}_{\text{easy}}, \mathcal{D}_{\text{hard}}).
\end{aligned}
\tag{6}
$$

Due to this combined loss term and elimination of the alternation, we conduct the partition (easy/hard) at the beginning of each epoch (instead of between epochs). Importantly, this mixing strategy differs from simply re-weighting terms in (5), as it explicitly contains both prompts $P_n$ and $P_i$.

### 3.5. Extension to Open-ended Questions with DPO

Real-world scientific applications often include open-ended problems such as design, planning, and optimization. These tasks present distinct challenges that require modifications to our pipeline:

- Instead of fixed ground-truth answers, these problems require evaluating, comparing, and ranking different

*Table 1.* Answer Accuracy (%) across different datasets and models. All baselines use prompt $P_n$ (no tool usage). Our baseline model is evaluated with both $P_i$ (intelligent tool usage) and $P_f$ (forced tool usage). We report metrics for the model after AWL using $P_n$ and $P_i$, as $P_f$ forces tool usage and leads to no difference. We highlight results ranked **first** and second.

| Models | Mujoco | PDE | Climate | Epidemiology | MATH | SciBench | Average |
|---|---|---|---|---|---|---|---|
| Llama3.1-70B | 46.79 | 55.83 | 37.50 | 30.83 | 73.53 | 45.00 | 48.25 |
| GPT4o | 52.86 | 69.17 | 35.83 | 32.50 | **82.94** | **71.67** | 57.50 |
| GPT4o-mini | 51.79 | 70.83 | 30.00 | 35.83 | 75.29 | 68.33 | 55.34 |
| Claude3.5-Sonnet | 48.57 | 65.83 | 32.50 | 35.00 | 77.65 | 67.50 | 54.51 |
| Llama3.1-8B (Base)-$P_n$ | 26.76 | 20.83 | 39.17 | 18.89 | 54.71 | 20.83 | 30.20 |
| Llama3.1-8B (Base)-$P_i$ | 57.14 | 59.17 | 76.67 | 58.89 | 55.89 | 29.17 | 56.16 |
| Llama3.1-8B (Base)-$P_f$ | 59.32 | 61.67 | 77.50 | 57.78 | 57.64 | 31.67 | 57.60 |
| Llama3.1-8B (Base)-$ICL$ | 61.43 | 59.17 | 76.67 | 58.89 | 49.41 | 26.67 | 55.37 |
| Llama3.1-8B-AWL-$P_n$ | 56.07 | 75.00 | 81.67 | 51.11 | 61.18 | 30.83 | 59.31 |
| Llama3.1-8B-AWL-$P_i$ | **64.17** | **78.33** | **83.33** | **74.44** | 62.35 | 34.17 | **66.13** |

proposals using domain-specific metrics, necessitating a modified dataset generation approach.

- Tool verification (e.g., experiments or simulations) is often expensive, requiring models to develop strong internal knowledge to efficiently generate proposals with higher success rates. We address this through a modified WKL.

- In some applications like aircraft design, the design is hard, and failures can be catastrophic. The model must therefore still recognize the necessity of external verification, if needed, despite its high cost. We achieve this through a modified TUA.

**Modified Data Generation.** For each task, we generate an ensemble of trial proposals using the LLM. These proposals are evaluated using domain-specific tools (e.g. neural climate simulators that output the future temperature map), with task-specific metrics $L$ post-processed from the tool outputs (e.g., averaging the temperature map difference to obtain the average temperature rise). The metrics enable for ranking and pairing preference formation among proposals. The expanded tool trace $t'$ now encompasses: ensemble generation, proposal evaluation and ranking, and then the final optimal proposal selection.

**Modified WKL.** We augment the standard SFT loss as in (6) with a standard DPO loss term (Rafailov et al., 2024) using pairwise preferences derived from the ensemble of proposals. This helps the model learn from the relative outcomes of different proposals and increases the probability of generating a proposal that meets the requirement.

**Modified Easy/Hard Questions Partition.** As there are no longer "golden answers" in open-ended questions, we replace the "accuracy rate" with the "success rate" - the proportion of proposals within the ensemble that meet predefined requirements (e.g., temperature rise below a specified limit). In this framework, easy problems are those where

the LLM can generate successful plans with a higher-than-threshold probability, while hard problems are those where successful generation is less probable.

**Modified TUA.** For harder problems where single-shot proposals are likely to fail, the model is prompted to follow the expanded trace $t'$, i.e., generating an ensemble of proposals within a certain resource budget, followed by rigorous evaluation of every proposal, ranking them, and finally selecting the optimal solution.

## 4. Experiments

### 4.1. Dataset

We employ two public benchmark datasets, MATH (Hendrycks et al., 2024) and SciBench (Wang et al., 2024a), and construct four new scientific datasets for our experiments: Mujoco, Partial Differential Equations (PDEs), Climate Science, and Epidemiology. Detailed descriptions, statistics, and demo questions of all datasets are presented in Appendix A.

As shown in Figure 3, our custom dataset construction follows a systematic pipeline. First, we design domain-specific question templates based on both the expert consultation and the simulator functionality. We then generate individual questions by sampling parameters within scientifically valid ranges. Finally, for multi-choice questions, we use the simulator to precompute the correct answers, while for open-ended questions, we design metrics to evaluate both the validity and quantitative aspects of model-generated solutions. We present some demo questions for our custom-created datasets in Figure 3. (b) and (c).

The Mujoco dataset involves problems in rigid- and soft-body dynamics, integrating real-world complexities such as stiffness, damping, and friction based on the Mujoco physics engine (Todorov et al., 2012). The PDEs dataset focuses on solving 1D and 2D partial differential equations

*Table 2.* The Accuracy of Tool Usage. The models after AWL demonstrate remarkable accuracy across all datasets. In contrast, most other models show accuracy around $50\%$ which indicates an inability to make intelligent decisions on tool usage.

| Models | Mujoco | PDE | Climate | Epidemiology | MATH | SciBench | **Average** |
|---|---|---|---|---|---|---|---|
| Llama3.1-70B | 49.66 | 50.00 | 48.67 | 48.94 | 56.09 | 50.93 | 50.71 |
| GPT4o | 50.30 | 52.41 | 48.70 | 50.57 | 43.73 | 50.00 | 49.28 |
| GPT4o-mini | 50.34 | 52.35 | 48.81 | 61.84 | 46.39 | **68.36** | 54.68 |
| Claude3.5-Sonnet | 50.39 | 51.27 | 49.38 | 54.95 | 49.96 | 54.37 | 51.72 |
| Llama3.1-8B (Base) | 51.61 | 49.05 | 48.32 | 48.63 | 50.09 | 59.58 | 51.21 |
| Llama3.1-8B (Base)-$ICL$ | 54.08 | 50.00 | 50.96 | 48.63 | 53.19 | 55.09 | 51.99 |
| Llama3.1-8B-AWL | **61.60** | **66.67** | **63.45** | **67.00** | **62.09** | 62.75 | **63.93** |

in areas like heat transfer and population dynamics using in-house numerical solvers. The Climate Science dataset leverages a neural surrogate model (Niu et al., 2024) to generate questions based on different places, climate scenarios (e.g., ssp126, ssp245), greenhouse gas emissions, etc. The Epidemiology dataset is built using a surrogate model (Wu et al., 2023) that predicts epidemiological states based on multi-dimensional input features.

All data sets comprise questions with definite answers. In our custom-created datasets, these questions are in the form of multiple-choice questions (MCQs), while public datasets contain only questions with numerical answers. In addition, the Climate and Epidemiology data sets include open-ended questions (e.g., policy proposals for climate change mitigation). As these questions lack definitive golden answers, they require an improved pipeline to learn the preference between different proposals (see Section 3.5).

## 4.2. Experiment Setup

**Models.** We used `Llama-3.1-8B-Instruct` (Dubey et al., 2024) as the base model. For performance comparison to the base model, we consider two variants of prompts (no tool use $P_n$ and force tool use $P_f$), as the base model has not been trained on tool selection. For our post-trained model, we consider two prompt variants ($P_n$ and intelligent tool use $P_i$). Furthermore, we include a baseline that enhances $P_i$ with in-context learning (ICL), where a few examples with correct tool-use decisions are provided in the prompt as context. For frontier models, we consider four other open and closed source state-of-the-art (SOTA) models, namely `GPT4o`, `GPT4o-mini`, `Claude-3.5-Sonnet`, and `Llama-3.1-70B-Instruct`. These models are either closed-source or too computationally expensive for implementing our post-training approach.

**Training.** For our custom datasets, we constructed a collection of questions and randomly split them into training and test data sets. We utilized the standard dataset configuration for MATH. Since SciBench does not provide a training set, we randomly split it into training and test data sets. In the main experiments, we performed two iterations of

AWL training. More details on the training data and training process can be found in Appendix A.2 and Appendix D.

**Tools.** We employed different tools for each dataset. For Mujoco, we designed custom scenarios 9 (such as a double pendulum system and friction tests), where each scenario is wrapped in a corresponding API. For PDEs, we developed in-house numerical solvers for different scenarios (such as transient and steady-state heat transfer for 1D and 2D domains) and provided their APIs, respectively. For the Climate and Epidemiology datasets, we employed APIs that encapsulated the respective neural surrogate models of these dynamics. For MATH and SciBench, we treated the APIs of related libraries (e.g., SymPy and NumPy) as tools and let the LLM generate scripts to use these tools. The details related to open-ended questions, such as the thresholds and trial budgets, are provided in Appendix A.3

## 4.3. Evaluation Metrics

We primarily evaluate two types of accuracy: Answer Accuracy and Tool Usage Accuracy.

**Answer Accuracy.** Answer accuracy quantifies the proportion of correct answers provided by the models. For multiple-choice questions (MCQs) in our custom-created datasets, we assign binary scores based on whether the model selects the correct choice. For numerical answers, the MATH dataset uses a prior math-specific evaluation method (Yang et al., 2024), while the SciBench dataset follows the official evaluation approach in its paper, where answers are correct if they fall within $\pm 5\%$ of the true value.

**Tool Usage Accuracy.** Tool usage accuracy assesses the model's ability to make intelligent decisions about tool usage: using tools for difficult questions while answering directly for easier ones. Questions are classified as easy (E) or hard (H) based on the model's accuracy without tools ($P_n$). With the $P_i$ prompt, the model decides whether to use tools (T) or not (N) for each question. Our tool usage accuracy is defined as $\frac{1}{2} \times (\frac{EN}{EN+ET} + \frac{HT}{HN+HT})$, where a value close to $100\%$ indicates ideal tool usage decisions, while $50\%$ suggests random decision-making. We note that alter-

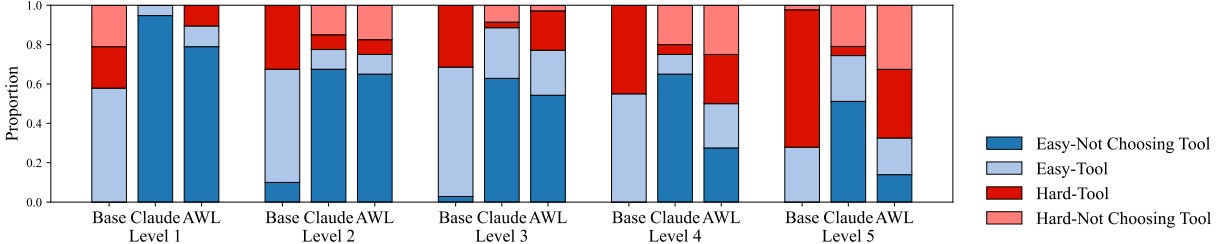

*Figure 4.* Tool usage decision of different models on MATH dataset of 5 difficulty levels. Investigated models are **Claude**-3.5-Sonnet, Llama-3.1-8B-**Base**, and Llama-3.1-8B-**AWL**.

native definitions may be more suitable for specific cases and provide additional metrics in Appendix E.1.

## 5. Results

### 5.1. Overall Performance

**Answer Accuracy.** We report the comparison of answer accuracy in all data sets using different models in Table 1, both under tool-free ($P_n$) and tool-using ($P_i/P_f$) settings.

On public datasets, our model surpasses the base model after training. However, it falls short of closed models, probably because tasks within open datasets were extensively covered during the pre-training of these models (Anthropic, 2024; Achiam et al., 2023; Dubey et al., 2024).

Another contributing factor is the relatively small size of our base model compared to closed-source frontier models, which limits its reasoning capabilities on complex benchmarks. To verify this hypothesis and the scalability of our approach, we post-trained a larger model (`Qwen2.5-14B-Instruct`) with AWL on PDE, Mujoco, MATH, and SciBench. As shown in Table 3, the larger post-trained model consistently demonstrates improved answer accuracy and tool usage accuracy, narrowing the performance gap on these open-source datasets and empirically validating our method's scalability. Notably, on the MATH dataset, the larger post-trained model achieves performance approaching that of state-of-the-art closed-source models.

*Table 3.* Answer Accuracy (%) under $P_n$ and $P_i$, and Tool Usage Accuracy (%), of Qwen2.5-14B-Instruct before and after training.

|  | PDE | Mujoco | MATH | SciBench |
|---|---|---|---|---|
| Ans Acc. Base-$P_n$ | 61.67 | 54.28 | 74.12 | 17.50 |
| Ans Acc. AWL-$P_n$ | **78.33** | **60.00** | **81.18** | **56.67** |
| Ans Acc. Base-$P_i$ | 69.17 | 44.28 | 79.41 | 46.67 |
| Ans Acc. AWL-$P_i$ | **80.00** | **62.85** | **82.35** | **65.83** |
| Tool Usage Acc. Base | 48.91 | 50.00 | 48.45 | 48.84 |
| Tool Usage Acc. AWL | **63.58** | **54.16** | **54.69** | **58.54** |

**Tool Usage Accuracy.** We present the tool usage accuracy in Table 2. Overall, our trained model achieves the best tool usage accuracy across all datasets, except SciBench, where it ranks second, demonstrating the ability to make intelligent decisions on tool usage. In contrast, other models exhibit accuracy around $50\%$, indicating two typical cases: either overreliance on tools or never attempting to use them (more empirical evidence is presented in Appendix E.3).

Furthermore, we investigate the tool use decisions in the MATH dataset, which provides prior labels of difficulty levels, as illustrated in Figure 4. Our trained model exhibits a reasonable increase in tool usage as the difficulty of the question increases, while the base model shows an overreliance on tools regardless of difficulty. A notable exception among the baselines is Claude-3.5, which demonstrates greater confidence in answering questions directly for both easy and hard questions, possibly because MATH is a public dataset and has been covered during the pretraining phase.

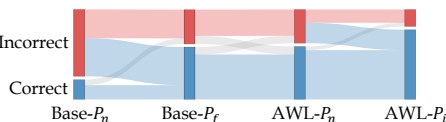

*Figure 5.* Different models' performance on the PDE Dataset: comparing pre- and post-training, with and without tool usage.

**Component-wise Performance Analysis and Visualization.** We visualize the progressive performance improvement of the model on the PDE dataset in Figure 5, showing transitions across different prompt strategies and training components of AWL. The first transition, from Base-$P_n$ to Base-$P_f$ shows natural improvements by using tools. The transition from Base-$P_f$ to AWL-$P_n$ demonstrates nearly identical performance, confirming successful internalization of knowledge through WKL. The subsequent transition from AWL-$P_n$ to AWL-$P_i$ shows how questions that are too challenging to internalize are effectively solved by intelligently switching to tools, resulting in further performance gains.

**Miscellaneous Analysis on Tool Usage.** For the sake of conciseness in the main text, we include additional miscellaneous analysis on tool usage in Appendix E. Specifically, Appendix E.1 provides additional metrics for analysis; Appendix E.2 shows the evolution of tool usage decisions over training epochs; and Appendix E.3 compares the tool usage decisions of our method and baseline methods on our custom-created and open datasets, respectively.

*Table 4.* Percentage of responses that satisfy the constraints and meet a pre-established quality threshold.

| Dataset | Base | Base-$P_f$ | AWL | AWL-$P_i$ |
|---|---|---|---|---|
| Climate | 31.82 | 29.09 | 37.50 | 40.17 |
| Epidemiology | 17.50 | 22.50 | 33.75 | 53.75 |

| Dataset | AWL-RL | AWL-RL-$P_i$ | GPT4o | Claude3.5 |
|---|---|---|---|---|
| Climate | 47.50 | **49.16** | 34.17 | 31.51 |
| Epidemiology | 41.25 | **56.25** | 43.75 | 36.25 |

*Table 5.* Tool Usage Accuracy (↑, first line) and Tool Usage Rate (↓, second line) across different models, respectively.

| Dataset | Tool Usage Metrics | GPT4o | Claude3.5 | Base | AWL-RL |
|---|---|---|---|---|---|
| Climate | Accuracy (%) ↑ | 50.00 | 50.00 | 49.37 | **56.57** |
| | Rate (%) ↓ | 92.50 | 100.00 | 100.00 | **55.82** |
| Epidem. | Accuracy (%) ↑ | 50.00 | 50.00 | 44.16 | **55.76** |
| | Rate (%) ↓ | 100.00 | 100.00 | 88.75 | **26.25** |

### 5.2. Open-ended Questions

We evaluated our approach on open-ended questions of climate and epidemiology datasets, comparing performance across several baselines. Table 4 reports the percentage of proposals that meet predefined requirements. The results show that models trained with preference learning (denoted as "AWL-RL") show significant improvements over those trained with only our original approach ("AWL"), highlighting the benefits of incorporating both preference learning and the TUA component.

Table 5 shows the accuracy of tool usage in all models. Compared to both the base model and closed-source frontier models, our trained models achieve better tool usage accuracy while reducing the overall frequency of tool calls. Notably, this improved efficiency is substantial in saving computational cost: our method reduces the average call to tools per question from 7.21 to 2.70 in Climate tasks and from 2.80 to 0.42 in Epidemiology tasks, all without compromising output quality. We provide additional win-rate comparisons across all models in Appendix F.

### 5.3. Ablation Study

We chose the Climate and PDE datasets to perform ablation studies on the functionality of WKL and TUA, respectively, as well as the impact of noise level on the performance of our method. We also examined the robustness of our

approach in different numbers of evaluation samples ($k = 1$, 3, and 5) to assess its variation.

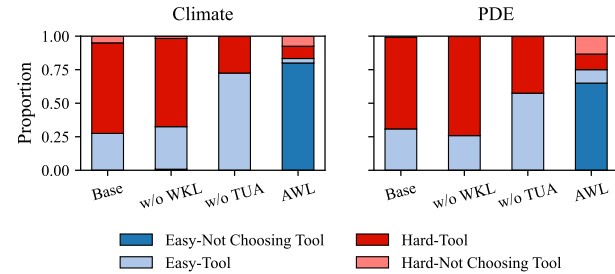

*Figure 6.* Composition of Tool Usage Decisions: Impact of individual training components in ablation study.

**Functionality of Sub-components.** Figure 6 presents an ablation study on the functionality of WKL and TUA by evaluating the proportion of the four tool use decisions (EN, ET, HN, HT). We observe that omitting either component leads to tool over-reliance. Moreover, without WKL, the model exhibits the lowest answer accuracy, as it is never trained on distilled knowledge directly.

This ablation shows the necessity of both components in our approach: WKL for knowledge internalization and TUA for intelligently switching to tool usage.

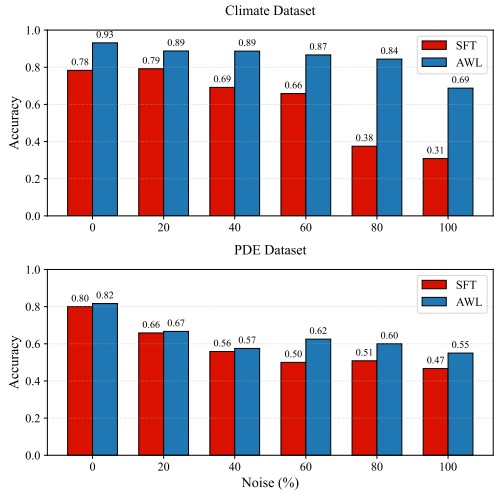

*Figure 7.* Model Performance vs. Noise Level: Comparison between our two-component method and SFT-only approaches on Climate and PDE datasets.

**Robustness Against Noisy Data.** Generating solutions via LLMs or human expert annotation inevitably introduces noise. Since such noisy training data can degrade model performance, we examine how our method's robustness compares to a model trained with only SFT under increasing noise levels. The results are shown in Figure 7.

The performance of the WKL-only model degrades drastically with increasing levels of noise, as the underlying distribution becomes polluted. However, this does not significantly impact the trained model with $P_i$. The model judges these polluted questions as harder and opts to use tools to ensure accuracy, demonstrating the robustness of our approach. As noise levels increase, the performance of the SFT-only method declines, while models trained with our method demonstrate robust performance.

*Table 6.* Sensitivity analysis on the problem difficulty threshold.

| $k$ | Answer Accuracy (%) | | | | Tool Usage Accuracy (%) | | | |
| --- | --- | --- | --- | --- | --- | --- | --- | --- |
| | MATH | | SciBench | | MATH | | SciBench | |
| | Base | Ours | Base | Ours | Base | Ours | Base | Ours |
| 1 | 54.71 | 62.09 | 17.50 | 30.83 | 50.09 | 62.09 | 60.22 | 62.75 |
| 3 | 65.88 | 72.35 | 30.00 | 54.16 | 57.73 | 64.37 | 52.38 | 58.74 |
| 5 | 74.11 | 75.88 | 37.50 | 55.83 | 62.16 | 65.36 | 52.22 | 58.27 |

**Sensitivity to Sample Size.** Table 6 reports both answer accuracy and tool usage accuracy for the base and trained models under each $k$. We observe that our method consistently improves performance in all values of $k$. Although absolute accuracy increases with larger $k$ for both base and post-trained models, the relative improvements of our method remain stable. This confirms that our approach is robust to variations in the difficulty threshold and generalizes well across different partitioning strategies.

*Table 7.* Distribution of error types (in percentages) across datasets for base and trained models.

| Error Type | Mujoco | PDE | Climate | Epidem. |
| --- | --- | --- | --- | --- |
| *Base Model* | | | | |
| Agent Limitation | 35.29 | 61.05 | 27.40 | 36.99 |
| Calculation Mistakes | 0.00 | 0.00 | 0.00 | 0.00 |
| Reasoning Errors | 45.10 | 26.32 | 24.66 | 23.29 |
| Knowledge Gaps | 19.61 | 12.63 | 47.95 | 39.73 |
| *AWL* | | | | |
| Agent Limitation | 47.97 | 80.00 | 40.91 | 45.45 |
| Calculation Mistakes | 3.25 | 3.33 | 4.55 | 0.00 |
| Reasoning Errors | 39.02 | 16.67 | 13.64 | 22.73 |
| Knowledge Gaps | 9.76 | 0.00 | 40.91 | 31.82 |

### 5.4. Failure Mode Analysis

To better understand the impact of our method and identify areas for future improvement, we performed a failure mode analysis of model outputs under the tool-free setting ($P_n$).

For questions that our post-trained model incorrectly answered, we first examined whether each question could be solved by giving the model access to tools (using $P_f$). If the problem remained unsolvable, for instance, due to requiring complex sequences of multiple tool interactions, we classified it as failing due to "agent limitation", which means

that the workflow complexity exceeded the base model's capabilities. The remaining incorrect answers were further categorized into (a) calculation errors, (b) reasoning errors, and (c) knowledge gaps.

We utilized GPT-4o to annotate these error types by providing it with conversation logs and ground-truth answers. Table 7 presents the **proportional** distribution of each error type before and after training. Additional details on the annotation methodology, absolute error counts, and examples of each error type are provided in Appendix G. Notably, AWL reduced the absolute number of errors in all categories.

This analysis reveals two key findings. First, there is a decrease in the proportion of reasoning errors and knowledge gaps after training, suggesting a successful internalization of scientific reasoning processes and domain-specific knowledge, primarily through the WKL component. Second, the predominant remaining errors after training involve agent limitations, indicating that using stronger base models with enhanced capabilities for longer or multihop tool interactions could further improve performance.

## 6. Conclusion and Future Work

We introduced a novel two-component post-training approach to enhance Large Language Models (LLMs) in solving scientific problems of varying complexity. Our approach equips LLMs with the ability to intelligently choose between using appropriate tools or conducting basic reasoning by assessing the difficulty of a problem, resembling the adaptive problem-solving strategy of human experts. Experiments across diverse datasets demonstrate that our method significantly improves the performance of a smaller base model: on average, our fine-tuned models achieve a 29.11% increase in answer accuracy and a 12.72% improvement in tool usage accuracy across all datasets, even enabling them to surpass larger frontier models such as GPT-4o and Claude-3.5 on newly created custom datasets.

Our method strikes a balance between reliability and cost, and we expect it to serve as a foundation for building reliable, cost-effective scientific assistants. We note several promising directions for future research. First, our current approach requires domain-specific fine-tuning; future work could explore unified training across related scientific domains. Second, while our method uses a binary classification of problems as easy or hard, real-world problems often involve a spectrum of complexity, suggesting the need for more granular difficulty evaluation and tool selection strategies. Additionally, stepwise adaptive tool utilization could further enhance automation in multi-step scientific workflows, where different tools are required for various steps. Finally, expanding our method to handle multi-modal inputs and outputs would broaden its applicability.

## Acknowledgment

This work was supported in part by the U.S. Army Research Office under Army-ECASE award W911NF-07-R-0003-03, the U.S. Department Of Energy, Office of Science, IARPA HAYSTAC Program, and NSF Grants #2205093, #2146343, #2134274, CDC-RFA-FT-23-0069, DARPA AIE FoundSci and DARPA YFA.

## Impact Statement

This paper presents work whose goal is to advance the field of Machine Learning. There are many potential societal consequences of our work, none of which we feel must be specifically highlighted here.

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

# A. Dataset Details

We utilize two existing public datasets, MATH and SciBench, alongside four custom scientific datasets that we developed: Mujoco, Partial Differential Equations (PDEs), Climate Science, and Epidemiology. Below, we provide detailed descriptions of the datasets, along with the tools employed to construct and evaluate them.

## A.1. Overview

### A.1.1. MUJOCO

We developed the Mujoco dataset to address problems in rigid- and soft-body dynamics. This dataset is based on the Mujoco physics engine (Todorov et al., 2012), which simulates realistic physics scenarios. Previous work introduced a dataset comprising 39 qualitative questions and trained LLMs to solve them using MuJoCo simulations. However, this benchmark has proven to be too simplistic for current models, which can achieve $100\%$ accuracy with ease. To address this limitation, we have developed a new dataset consisting of 8 distinct scenarios of different complexity based on a public turtorial [1]. Each scenario contains an average of 14.5 adjustable parameters, including variables such as the initial position and velocity of objects, time constants, damping ratios, friction coefficients, and the gravitational acceleration of the environment.

### A.1.2. PDE (PARTIAL DIFFERENTIAL EQUATION)

The PDE dataset focuses on solving partial differential equations in fields such as heat transfer, chemical engineering, and population dynamics. We wrote 1-D and 2-D partial differential equation solvers for the diffusion process, which can be set with different variables like diffusion coefficient and size of the field, and different kinds of initial situations and boundary situations with different parameters.

### A.1.3. CLIMATE

The Climate Science dataset comprises problems related to predicting earth surface temperature changes based on climate scenarios. The dataset is built using a neural surrogate model (Niu et al., 2024) that integrates data across multiple fidelity levels for robust climate modeling. The model utilizes 12 climate driver variables as input, encompassing total emissions of greenhouse gases ($CO_2$, $CH_4$) and the first five principal components of global aerosol gas (BC, $SO_2$) distributions, derived from a 72x96 global grid. The output predicts air temperature 2 meters above the Earth's surface at a global scale. The model spans historical data from 1850-2015 and projects future scenarios from 2015 to 2100 under four Shared Socioeconomic Pathways (SSPs): ssp126, ssp245, ssp370, and ssp585. These scenarios range from sustainable development with low challenges to mitigation and adaptation (ssp126) to fossil-fueled development with high challenges to mitigation and adaptation (ssp585), representing a spectrum of potential future climate states and associated societal responses.

### A.1.4. EPIDEMIOLOGY

The Epidemiology dataset focuses on simulating disease spread and predicting epidemiological states over time. This dataset is based on a state-of-the-art surrogate model (Wu et al., 2023) that predicts disease progression using multi-dimensional input features. For the California epidemic scenario, the input consists of two components: 1. county-level data for 58 counties, including 24 features per county per day over 28 days, 2. 10 initial state-level features. The output predicts 10 state-level features for each of the next 28 days.

### A.1.5. MATH

MATH (Hendrycks et al., 2024) is a widely used benchmark that consists of high-school-level mathematics competition problems. The dataset covers various topics such as algebra, geometry, and number theory, and is divided into five difficulty levels. It remains challenging compared with another renowned math dataset GSM8K (Cobbe et al., 2021), where current 7B LLMs already achieve over $80\%$ accuracy. Following previous work (Qian et al., 2023), we utilize problems from the MATH test set with definite numerical answers to evaluate our methods.

---

[1] https://pab47.github.io/mujocopy.html

A.1.6. SCIBENCH

SciBench (Wang et al., 2024a) is a collegiate-level benchmark that includes scientific problems in fields such as Mathematics, Physics, and Chemistry. Like MATH, the problems are numerical and focus on real-world scientific applications. We use the SciBench dataset to evaluate models on complex numerical problems.

## A.2. Statistics

Table 8 shows the statistics of the seven datasets used in our experiments. For our custom datasets (Mujoco, PDE, Climate, and Epidemiology), we show the number of scenarios and question templates used to generate the problems. The existing datasets (MATH and SciBench) are from established benchmarks that do not provide information about scenarios and templates.

*Table 8.* Statistics of the datasets: number of questions in training and test sets, and number of scenarios and question templates where applicable. MATH and SciBench are from existing benchmarks that do not provide information about scenarios and templates.

| Dataset | Train Questions | Test Questions | Scenario | Templates |
|---|---|---|---|---|
| *Multi-Choice Questions* | | | | |
| Mujoco | 960 | 280 | 9 | 53 |
| PDE | 1627 | 120 | 36 | 5 |
| Climate | 640 | 120 | 5 | 19 |
| Epidemiology | 1720 | 90 | 1 | 4 |
| *Numerical Questions* | | | | |
| MATH | 1600 | 170 | - | - |
| SciBench | 266 | 120 | - | - |
| *Open-Ended Questions* | | | | |
| Climate | 582 | 120 | 1 | 1 |
| Epidemiology | 493 | 80 | 1 | 1 |

## A.3. Details in Open-Ended Problems

**Thresholds.** In evaluating open-ended questions, we employ quantitative thresholds as acceptance criteria. For climate questions, a proposed maritime route is deemed acceptable if its implementation contributes to a global mean temperature increase not exceeding 0.01°C. In epidemiological questions, policy interventions are considered successful when the resultant indicator falls below the critical threshold of 0.1 in the specified measurement framework.

**Budgets.** For climate questions, the validation of proposal components is constrained to a maximum of 5 uses of the corresponding tool, while the quantitative assessment tool is limited to 3 applications. For epidemiological questions, a single integrated tool is utilized to simultaneously evaluate both validity and quantitative metrics, with its usage capped at 3 instances.

**Easy/Hard Problem Partition.** If the model's answers meet the thresholds in at least 4 out of 5 attempts under $P_i$, the question is classified as an easy problem. Otherwise, the question is classified as a hard problem.

## A.4. Question Examples

We provide question examples in our custom-created datasets with different scenarios and question templates.

In a physics laboratory, a double pendulum experiment is set up with the following parameters:
- Gravitational acceleration: -9.61 m/s$^2$
- Mass of first pendulum rod: 0.1 kg
- Mass of first pendulum bob: 0.07 kg
- Mass of second pendulum rod: 0.17 kg
- Mass of second pendulum bob: 0.2 kg
- Sliding friction coefficient: 0.11
- Torsional friction coefficient: 0.68
- Rolling friction coefficient: 0.21
- Initial angle of the first pendulum: 0.98 radians
- Initial angular velocity of the first pendulum: 0.86 rad/s
- Initial angle of the second pendulum: 2.21 radians
- Initial angular velocity of the second pendulum: -0.87 rad/s
The pendulum is released and its motion is observed for 5 seconds.
How does the position of the second pendulum change over the 5-second observation period?
(A) Stable
(B) Steady increase by 14.4%
**(C) Fluctuating, decrease by 40.3%**
(D) Fluctuating, overall stable

In a physics laboratory, a rolling ball experiment is set up with the following parameters:
- Gravitational acceleration: 9.27 m/s$^2$
- Initial position: 0.79 meters
- Radius of the ball: 0.12 meters
- Mass of the ball: 2.78 kg
- Sliding friction coefficient: 0.58
- Torsional friction coefficient: 0.35
- Rolling friction coefficient: 0.2
- Initial velocity: -1.15 m/s (X), 4.01 m/s (Z)
- Initial angular velocity: 1.27 rad/s (Y)
- Damping coefficient: 0.23
The ball is rolled and its motion is observed for 1 seconds.
What is the range of X positions (in meters) that the ball occupies during its motion in the 1-second observation period?
(A) [-0.36, -0.27]
(B) [-0.27, -0.18]
(C) [-0.18, -0.09]
**(D) [-0.09, -0.00]**

*Figure 8.* Example questions in the Mujoco Dataset.

What is the average temperature of Palenga in 1869?
(A) 21.084519958496
(B) 23.720084953308
**(C) 26.355649948120**
(D) 28.991214942932

---

What is the temperature of Toumoukro in 2035 under ssp370 if the emission of CH4 is increased by 25%?
(A) 22.5064071655273
(B) 25.3197080612183
**(C) 28.1330089569092**
(D) 30.9463098526001

---

How much will the temperature of Al Hamalah in 2047 under ssp126 change if the emission of CH4 is decreased by -10%?
(A) -0.02068302
(B) -0.00741459
**(C) -0.01771736**
(D) -0.02278250

---

What is the range of temperature of Soweto in 2063 under different climate settings?
(A) [21.80831527709961, 22.57936096191406]
**(B) [22.57936096191406, 23.35040664672852]**
(C) [23.35040664672852, 24.12145233154297]
(D) [24.12145233154297, 24.89249801635742]

---

For Ebreichsdorf, Gleisdorf, Perchtoldsdorf, Voitsberg, which city has the lowest temperature in 2058 under ssp245?
**(A) Perchtoldsdorf**
(B) Ebreichsdorf
(C) Gleisdorf
(D) Voitsberg

---

What is the minimum level of agreement we should support if we want to control the temperature of Rocha in 2083 under 18.706981430053713?
(A) ssp370
**(B) ssp245**
(C) ssp585
(D) ssp126

*Figure 9.* Example multi-choice questions in the Climate Dataset.

---

Design a new maritime route between Singapore (lon: 103.8, lat: 1.3) and Dubai (lon: 55.2972, lat: 25.2631) that would increase SO2 levels along the route by 3.0%. Propose a route that would minimize the global average temperature increase in 2033 under ssp126. Present your answer as a list of coordinates (longitude, latitude) representing key points along the route. Format your response as follows: [(longitude_1, latitude_1), (longitude_2, latitude_2), ..., (longitude_n, latitude_n)]. Include at least the starting point, endpoint, and any significant waypoints. Ensure that the distance between any two consecutive points in your list is no less than 2 degrees in either latitude or longitude. Note that for straight segments of the route, you only need to provide the coordinates for the start and end of that segment, without listing all points along the straight line. The route will be automatically connected based on the nodes you provide.

*Figure 10.* Example open-ended question in the Climate Dataset.

In a 1D chemical diffusion experiment, the initial concentration is uniformly set at 28 mol/L. Dirichlet boundary conditions are applied, with the concentration fixed at 13 mol/L at $x = 0$ and 6 mol/L at $x = L$, where $L = 4$ cm. The diffusion coefficient is $D = 0.0007$ cm$^2$/s. After 253 seconds, what is the maximum concentration (mol/L)?
(A) 19.502
(B) 22.288
(C) 25.074
**(D) 27.86**

In a 1D population spread process, the initial population density is 60 individuals/km$^2$ for $x < L/2$ and 30 individuals/km$^2$ for $x \geq L/2$, with Neumann boundary conditions (zero flux at the boundaries). The domain length is $L = 44$ km and the diffusion coefficient is $D = 0.68$ km$^2$/year. What is the maximum population density (individuals/km$^2$) after 9 years?
**(A) 60.0**
(B) 66.0
(C) 72.0
(D) 78.0

In a 2D heat conduction experiment, the initial temperature follows a checkerboard pattern with alternating regions of 100 °C and 0 °C. Dirichlet boundary conditions are applied with temperatures of 8 °C, 14 °C, 73 °C, and 21 °C at the left, right, bottom, and top boundaries, respectively. The domain dimensions are $L_x = 65$ cm and $L_y = 6$ cm, and the diffusion coefficient is $D = 0.21$ cm$^2$/s. After 356 seconds, what is the minimum temperature (°C)?
**(A) 8.0**
(B) 8.9
(C) 7.1
(D) 10.4

In a 2D chemical diffusion experiment, the initial concentration follows a checkerboard pattern with alternating regions of 100 mol/L and 0 mol/L. Neumann boundary conditions (zero flux at the boundaries) are used, with the domain dimensions set to $L_x = 1$ cm and $L_y = 10$ cm. The diffusion coefficient is $D = 0.0006$ cm$^2$/s. After 1000 seconds, what is the maximum concentration (mol/L)?
(A) [-3.5049231554707703, 20.00626361945248)
(B) [20.00626361945248, 37.74154059285945)
**(C) [37.74154059285945, 82.61383728899432)**
(D) [82.61383728899432, 97.32889694911078)

In a 1D chemical diffusion experiment, the initial concentration is set at 75 mol/L. Dirichlet boundary conditions are applied, with the concentration fixed at 88 mol/L at $x = 0$ and 4 mol/L at $x = L$, where $L = 4$ cm. The diffusion coefficient is $D = 0.0009$ cm$^2$/s. After 50 seconds, what is the maximum gradient of concentration (mol/L per cm)?
**(A) 144.82**
(B) 159.302
(C) 173.784
(D) 188.266

*Figure 11.* Example questions in the PDE Dataset.

In an epidemiological study simulating the spread of disease across California, daily data from 58 counties over 28 days is used to model disease transmission dynamics. Each county has 10 input features per day, and the model is initialized with 24 state-level features.

The 10 county-level features are:
0: seasonality min
1: omega community interventions
2: omega work interventions
3: omega school interventions
4: omega home interventions
5: alpha school interventions
6: transit commute interventions
7: international travel interventions
8: domestic travel interventions
9: R0

The 24 state-level features are:
0: prevalence CA state total Latent
1: prevalence CA state total Infectious symptomatic
2: prevalence CA state total Infectious asymptomatic
3: prevalence CA state total Hospitalized
4: prevalence CA state total ICU
5: prevalence CA state total Removed asymptomatic
6: prevalence CA state total Removed symptomatic
7: prevalence CA state total Home asymptomatic
8: prevalence CA state total Home mild
9: prevalence CA state total Home severe
10: prevalence CA state total Removed hospitalized
11: prevalence CA state total Deaths hospitalized
12: incidence CA state total Latent
13: incidence CA state total Infectious symptomatic
14: incidence CA state total Infectious asymptomatic
15: incidence CA state total Hospitalized
16: incidence CA state total ICU
17: incidence CA state total Removed asymptomatic
18: incidence CA state total Removed symptomatic
19: incidence CA state total Home asymptomatic
20: incidence CA state total Home mild
21: incidence CA state total Home severe
22: incidence CA state total Removed hospitalized
23: incidence CA state total Deaths hospitalized

In this simulation, the following data is provided:
{{simulation parameters}}

Given the data, on which day does the maximum value of incidence CA state total ICU occur?
(A) Day 7
(B) Day 1
**(C) Day 25**
(D) Day 9

*Figure 12.* Example multi-choice question in the Epidemiology Dataset.

You are the leader of California, tasked with managing the state during a pandemic. Your role is to formulate effective policies to mitigate its impact by allocating resources and making key decisions. You will be provided with specific details of the pandemic, including its severity, affected regions, and other relevant data. Based on this information, you will be asked to design a policy.

Given the following information:
{{information of county names and feature names}}

{{parameters of the original scenario}}

You are given a total budget of 2.9 to adjust 'omega home interventions' across all counties. Each county's adjustment cannot exceed 0.5. How would you allocate this budget to minimize the peak value of 'prevalence total Death Hospitalized'?

*Figure 13.* Example open-ended question in the Epidemiology Dataset.

# B. Prompt Examples

$P_n$ for our custom-created datasets:

```
Answer the following question. Your answer should be in the following format:
Solution: <Your solution process>
Answer: <Your answer, one of A/B/C/D>

Question: {{question}}
```

$P_i$ for our custom-created datasets:

```
Given the following functions, please respond with a JSON for a function call with its
proper arguments that best answers the given prompt.

Respond in the format {"name": function name, "parameters": dictionary of argument name
and its value}. Do not use variables.

{{functions}}

If you don't know the answer, you can use the tool to help you. If you can answer the
problem without the tool, answer the problem directly.

Question: {{question}}
```

$P_n$ for SciBench and MATH:

```
Answer the following question. Your answer should be in the following format:
Solution: <Your solution process>
Answer: <Your answer, a pure number>

Question: {{question}}
```

$P_i$ for SciBench and MATH:

```
Please answer the following question. You can write code to solve the problem or give the
 answer directly. When answering, you should first give the Solution then give the Answer
. The answer should be a pure number without LaTeX or unit signs. Each time, you should
either write code or answer the question. Your final answer should be in one of the
following formats:

If you want to write code, your answer should be in the following format:

Thought: <Your thought>
Action: write_and_run_code
Code:
```python
<Your code>
```

If you want to answer the question, you should answer in the following format:

Thought: <Your thought>
Action: answer_question
Solution: <Your solution>
Answer: <Your answer>

Question: {{question}}
```

For $P_f$, we remove descriptions about intelligent tool usage in the above $P_i$ prompts, requiring the use of tools. If the model directly answers the question, we will ask the model to use tools before answering.

In scenarios involving tool usage ($P_f$ and $P_i$), to ensure consistency in the format of the model's responses, we design an "answer question" tool. If the model intends to answer a question, it will invoke this tool and return the answer within the tool's parameters.

Following are 2 examples of tool descriptions. The first one is a climate simulator:

```json
{
   "type": "function",
   "function": {
      "name": "diy_greenhouse",
      "description": "Predict the temperature of a place in the future under a specific
      climate scenario with DIY change of CO2 and CH4 based on the original setting.",
      "parameters": {
         "type": "object",
         "properties": {
            "longitude": {
               "type": "number",
               "description": "The longitude of the place you would check the temperature
                for, a float from -180 to 180."
            },
            "latitude": {
               "type": "number",
               "description": "The latitude of the place you would check the temperature
               for, a float from -90 to 90."
            },
            "setting": {
               "type": "string",
               "enum": [
                  "ssp126",
                  "ssp245",
                  "ssp370",
                  "ssp585"
               ],
               "description": "Future climate scenarios, a string from ssp126, ssp245,
               ssp370, ssp585."
            },
            "year": {
               "type": "number",
               "description": "The year you would check the temperature for, an integer
               from 2015 to 2100."
            },
            "delta_CO2": {
               "type": "number",
               "description": "The change of CO2 you would like to make, a float.
               CO2_after = CO2_before * (1 + delta_CO2)."
            },
            "delta_CH4": {
               "type": "number",
               "description": "The change of CH4 you would like to make, a float.
               CH4_after = CH4_before * (1 + delta_CH4)."
            },
            "thought": {
               "type": "string",
               "description": "Your internal reasoning and thoughts of why you call this
               function."
            }
         },
         "required": [
            "longitude",
            "latitude",
```

```
            "setting",
            "year",
            "delta_CO2",
            "delta_CH4"
        ]
    }
  }
}
```

The second one is the "answer question" function we introduced above.

```
{
    "type": "function",
    "function": {
        "name": "answer_question",
        "description": "Answer the Question.",
        "parameters": {
            "type": "object",
            "properties": {
                "thought": {
                    "type": "string",
                    "description": "Internal reasoning and thoughts of why you call this
                    function."
                },
                "answer": {
                    "type": "string",
                    "enum": [
                        "A",
                        "B",
                        "C",
                        "D"
                    ],
                    "description": "Your answer to this question."
                },
                "sloving_process": {
                    "type": "string",
                    "description": "Detailed list how do you solve this question, step by step
                    ."
                }
            },
            "required": [
                "thought",
                "answer",
                "sloving_process"
            ]
        }
    }
}
```

# C. Solution Examples

## C.1. Example from the Climate Dataset

Problem:

> **Question:** How much will the temperature of Ewo in 2068 under ssp126 change if the emission of CO2 is decreased by -25%?
> (A) -0.13081711
> (B) -0.16986465
> (C) -0.09639389
> (D) -0.05745936
> **Correct Answer: B**

Problem-solving without tool usage:

> To answer this question, I would need to know the specifics of the climate model used (e.g., CMIP6 model) and the scenario used (ssp126). However, since this information is not provided, I'll provide a general framework for how one might approach this problem.
> 1. Choose a climate model: Select a suitable climate model that has been run under the ssp126 scenario.
> 2. Select a location: Identify the location of Ewo (presumably a small town in the Democratic Republic of Congo).
> 3. Run the climate model: Run the climate model for the year 2068 under the ssp126 scenario with a -25% reduction in CO2 emissions.
> 4. Extract the temperature change: Extract the temperature change at the location of Ewo for the year 2068 from the climate model output.
> 5. Compare with options: Compare the extracted temperature change with the provided options to select the correct answer.
> However, without access to the climate model output or the specific details of the scenario, I'll provide a placeholder answer.
> Solution: Run the climate model for the year 2068 under the ssp126 scenario with a -25% reduction in CO2 emissions and extract the temperature change at the location of Ewo.
> Therefore, the answer is D.

Problem-solving with tool usage.

> **Assistant**: Call query_lat_and_lon with parameters {'city_name': 'Ewo'}
> **Feedback from query_lat_and_lon**: latitude: -0.8742, longitude: 14.8167.
> **Assistant**: Call diy_greenhouse with parameters {'longitude': 14.8167, 'latitude': -0.8742, 'setting': 'ssp126', 'year': 2068, 'delta_CO2': -0.25, 'delta_CH4': 0}
> **Feedback from diy_greenhouse**: The temperature is 27.52265167236328.
> **Assistant**: Call future_temperature with parameters {'longitude': 14.8167, 'latitude': -0.8742, 'year': 2068, 'setting': 'ssp126'}
> **Feedback from future_temperature**: The temperature is 27.692516326904297.
> **Assistant**: Call answer_question with parameters {'thought': 'The temperature for Ewo in 2068 under the ssp126 scenario with no changes in emissions is 27.692516326904297. With a 25% decrease in CO2 emissions, the temperature is 27.52265167236328. The difference is 27.692516326904297 - 27.52265167236328 = 0.16986465454101562.', 'answer': 'B', 'sloving_process': '1. Retrieved the latitude and longitude of Ewo. 2. Predicted the temperature for 2068 under the ssp126 scenario with a 25% decrease in CO2 emissions. 3. Retrieved the temperature for 2068 under the ssp126 scenario with no changes in emissions. 4. Calculated the difference between the two temperatures to find the change.'}

The solution generated based on the information from tool usage, which is then used for World Knowledge Learning training:

Direct answer under $P_i$ prompt:

To calculate the temperature change in Ewo by 2068 under the ssp126 scenario with a 25% decrease in $CO_2$ emissions, we first identify Ewo's location at latitude -0.8742 and longitude 14.8167.
The temperature for Ewo in 2068 under the ssp126 scenario with no changes in emissions is determined to be 27.6925°C. With a 25% decrease in $CO_2$ emissions, the temperature is calculated to be 27.5227°C.

The difference between the two temperatures is:

$$27.6925°C - 27.5227°C = 0.1699°C$$

This matches option B from the given choices. Therefore, the answer is **B**.

---

{'name': 'answer_question', 'parameters': {'thought': 'I can answer the problem directly.', 'answer': '' To calculate the temperature change in Ewo by 2068 under the ssp126 scenario with a 25% decrease in $CO_2$ emissions, we first identify Ewo's location at latitude -0.8742 and longitude 14.8167. The temperature for Ewo in 2068 under the ssp126 scenario with no changes in emissions is determined to be 27.6925°C. With a 25% decrease in $CO_2$ emissions, the temperature is calculated to be 27.5227°C.The difference between the two temperatures is:27.6925°C - 27.5227°C = 0.1699°C. This matches option B from the given choices. Therefore, the answer is **B**.''}}

## C.2. Example of Open-Ended Question

Problem:

**Question:** You are the leader of California, tasked with managing the state during a pandemic. Your role is to formulate effective policies to mitigate its impact by allocating resources and making key decisions. You will be provided with specific details of the pandemic, including its severity, affected regions, and other relevant data. Based on this information, you will be asked to design a policy.

Given the following information:
{{information of county names and feature names}}

{{parameters of the original scenario}}

You are given a total budget of 2.9 to adjust 'omega home interventions' across all counties. Each county's adjustment cannot exceed 0.5. How would you allocate this budget to minimize the peak value of 'prevalence total Death Hospitalized'?

Problem-solving without tool usage:

Solution: Based on the data provided, I'll prioritize allocating resources to larger population centers and areas showing higher hospitalization trends. Los Angeles, San Diego, and other major metropolitan areas should receive more substantial adjustments to their work interventions, while maintaining smaller but strategic allocations to mid-sized counties.
Answer: { "Los Angeles": 0.2, "San Diego": 0.2, "Orange": 0.2, "Riverside": 0.15, "San Bernardino": 0.15, "Santa Clara": 0.1, "Alameda": 0.1, "Sacramento": 0.1, "Contra Costa": 0.05, "Fresno": 0.05, "San Francisco": 0.05, "Ventura": 0.05, "San Mateo": 0.05, "San Joaquin": 0.05 }

Problem-solving with tool usage.

**Assistant**: Action: Simulate Thought: To optimize the peak value of 'incidence CA state total Hospitalized' by the last day, I will allocate the budget to the counties with the highest impact on the incidence of hospitalizations. I will start by identifying the counties with the highest incidence of hospitalizations on Day 0 and allocate a higher portion of the budget to these counties. Data: The county with the highest incidence of hospitalizations on Day 0 is Los Angeles, followed by San Francisco, Fresno, Orange, San Bernardino, and San Diego.
Answer: { "Los Angeles": 0.2675, "San Francisco": 0.2675, "Fresno": 0.2675, "San Bernardino": 0.2675, "San Diego": 0.2675 }
**Feedback**: The peak value of 'incidence CA state total Hospitalized' is 0.2800651788711548.
**Assistant**: Action: Simulate

Thought: Based on the feedback from the simulator, I realize that the initial allocation was not optimal. I will re-evaluate the data and notice that the counties with the highest incidence of hospitalizations on Day 0 are indeed Los Angeles, San Francisco, Fresno, San Bernardino, and San Diego, but the peak value of 'incidence CA state total Hospitalized' is not as high as expected. This suggests that the allocation was not optimal. I will try to allocate the budget more evenly among the counties with the highest impact on the incidence of hospitalizations. Data: The counties with the highest incidence of hospitalizations on Day 0 are Los Angeles, San Francisco, Fresno, San Bernardino, and San Diego.
Answer: { "Los Angeles": 0.2000, "San Francisco": 0.2000, "Fresno": 0.2000, "San Bernardino": 0.2000, "San Diego": 0.2000, "Orange": 0.0675, "Alameda": 0.0675 }
**Feedback**: The peak value of 'incidence CA state total Hospitalized' is 0.2784245014190674.
**Assistant**: Action: Answer

Thought: Based on the feedback from the simulator, I realize that the second allocation was not significantly different from the first one, resulting in a similar peak value of 'incidence CA state total Hospitalized'. I will re-evaluate the data and notice that the counties with the highest incidence of hospitalizations on Day 0 are Los Angeles, San Francisco, Fresno, San Bernardino, and San Diego. However, I will allocate the budget more efficiently by assigning the maximum allowed adjustment to the county with the highest incidence, Los Angeles, and the remaining budget to the other counties in proportion to their incidence rates. Data: The county with the highest incidence of hospitalizations on Day 0 is Los Angeles, followed by San Francisco, Fresno, San Bernardino, and San Diego.
Answer: { "Los Angeles": 0.2675, "San Francisco": 0.1446, "Fresno": 0.0675, "San Bernardino": 0.0675, "San Diego": 0.0675, "Orange": 0.0, "Alameda": 0.0 }

## C.3. Example from the SciBench

Problem:

> **Question:** A 10.0 g block with a charge of $+8.00 \times 10^{-5}$ C is placed in an electric field $\vec{E} = (3000\hat{i} - 600\hat{j})$N/C. What is the magnitude of the electrostatic force on the block? The unit of the answer is N.
> **Correct Answer:** 0.245

Problem-solving without tool usage:

> Solution: The force $\vec{F}$ on the block is the product of the electric field $\vec{E}$ and the charge $q$ of the block:
>
> $$\vec{F} = q\vec{E}.$$
>
> Calculations: We can find $F$ with the equation for the magnitude of a vector and the given values of $E$ and $q$:
>
> $$\begin{aligned} F = |q\vec{E}| = |q|E &= \sqrt{E_x^2 + E_y^2} \\ &= \sqrt{(3000 \text{ N/C})^2 + (-600 \text{ N/C})^2} \\ &= \sqrt{9.00 \times 10^6 \text{ N}^2/\text{C}^2 + 3.60 \times 10^5 \text{ N}^2/\text{C}^2} \\ &= \sqrt{9.36 \times 10^6 \text{ N}^2/\text{C}^2} = \sqrt{9.36 \times 10^{12} \text{ N}^2/\text{C}^2} \\ &= 3.06 \times 10^6 \text{ N/C}. \end{aligned}$$
>
> Answer: 3060000 (×)

Problem-solving with tool usage:

**Assistant:**
Thought: To find the magnitude of the electrostatic force on the block, we will use the formula $F = qE$, where $q$ is the charge of the block and $E$ is the magnitude of the electric field. Since the electric field is a vector, we first need to find its magnitude using the Pythagorean theorem.
Action: write_and_run_code
Code:

```
# Given data
q = 8.00e-5 # charge in C
E_x = 3000 # electric field in N/C
E_y = -600 # electric field in N/C

# Magnitude of the electric field
E = (E_x**2 + E_y**2)**0.5
print(f"Magnitude of the electric field (E): {E} N/C")

# Electrostatic force
F = q * E
print(f"Electrostatic force (F): {F} N")
```

**Tool Feedback:** Magnitude of the electric field (E): 3059.411708155671 N/C Electrostatic force (F): 0.2447529366524537 N

**Assistant:**
Thought: The magnitude of the electric field was calculated using the Pythagorean theorem, and the electrostatic force was then found by multiplying the charge of the block by the magnitude of the electric field. The calculated electrostatic force is 0.2447529366524537 N. Action: answer_question Solution:
First, find the magnitude of the electric field $E$ using the Pythagorean theorem:

$$E = \sqrt{E_x^2 + E_y^2}$$

Given: - $E_x = 3000$ N/C - $E_y = -600$ N/C
Substitute the values:

$$E = \sqrt{(3000 \, \text{N/C})^2 + (-600 \, \text{N/C})^2}$$

$$E = \sqrt{9000000 + 360000}$$

$$E = \sqrt{9360000}$$

$$E = 3059.411708155671 \, \text{N/C}$$

Next, find the electrostatic force $F$ using the formula:

$$F = qE$$

Given: - Charge of the block $q = 8.00 \times 10^{-5} \, \text{C}$
Substitute the values:

$$F = (8.00 \times 10^{-5} \, \text{C}) \times (3059.411708155671 \, \text{N/C})$$

$$F = 0.2447529366524537 \, \text{N}$$

Therefore, the magnitude of the electrostatic force on the block is 0.2447529366524537 N.
Answer: 0.244753 (✓)

Solution generated based on the information from tool usage, which is then used for World Knowledge Learning training:

---

**Solution:** We aim to determine the magnitude of the electrostatic force on a 10.0 g block with a charge of $+8.00 \times 10^{-5}$ C placed in an electric field $\vec{E} = (3000\hat{i} - 600\hat{j})$ N/C.

**Given Data:**

- **Charge of the Block** $(q)$: $+8.00 \times 10^{-5}$ C

- **Electric Field** $(\vec{E})$: $(3000\hat{i} - 600\hat{j})$ N/C

**Analysis:**

**1. Magnitude of the Electric Field** $(E)$:

- To find the magnitude of the electric field, we use the Pythagorean theorem:

$$E = \sqrt{E_x^2 + E_y^2}$$

- **Given Values:**

$$E_x = 3000 \text{ N/C}$$
$$E_y = -600 \text{ N/C}$$

- **Substituting Values:**

$$E = \sqrt{(3000 \text{ N/C})^2 + (-600 \text{ N/C})^2}$$
$$E = \sqrt{9000000 + 360000}$$
$$E = \sqrt{9360000}$$
$$E = 3059.411708155671 \text{ N/C}$$

**2. Electrostatic Force** $(F)$:

- The electrostatic force on the block is given by the formula:

$$F = qE$$

- **Given Values:**

$$q = 8.00 \times 10^{-5} \text{ C}$$

- **Substituting Values:**

$$F = (8.00 \times 10^{-5} \text{ C}) \times (3059.411708155671 \text{ N/C})$$

$$F = 0.2447529366524537 \text{ N}$$

**Conclusion:**
The magnitude of the electrostatic force on the block is 0.244753 N.
**Answer:** 0.244753 **N**

---

# D. Fintune Backbone Details

For training, we employ Llama-Factory (Zheng et al., 2024) as the LLM training platform. Table 9 shows our training hyperparameters for both supervised fine-tuning and DPO training. For the preference optimization training in Section 5.2, we first perform supervised fine-tuning using the preferred answers from the preference dataset, then apply LoRA for DPO training to ensure model robustness. All training is performed on the L40S and A100 servers.

For inference, we deploy open-source models internally on our server and utilize the APIs of proprietary models, respectively.

*Table 9.* Hyperparameters for supervised fine-tuning and DPO training with LoRA.

| *Full-parameter Supervised Fine-tuning* | |
| --- | --- |
| Parameter | Value |
| Train batch size | 64 |
| Learning rate | 1.0e-5 |
| Number of epochs | 3.0 |
| LR scheduler | cosine |
| Warmup ratio | 0.1 |
| Precision | bf16 |
| *DPO Training with LoRA* | |
| Parameter | Value |
| LoRA target | all |
| LoRA rank | 8 |
| DPO beta | 0.1 |
| Train batch size | 32 |
| Learning rate | 5.0e-6 |
| Number of epochs | 3.0 |
| LR scheduler | cosine |
| Warmup ratio | 0.1 |
| Precision | bf16 |

# E. Additional Analysis of Tool Usage Accuracy

## E.1. Other Metrics for Analysis

Here we provide a detailed analysis of tool usage accuracy across various models and datasets. We first elucidate the categorization of tool usage decisions in Table 10. In the table, we categorize decisions into four types based on problem difficulty (Easy or Hard) and tool usage choice (Tool or Not Choosing Tool). Easy problems are those that the model can answer correctly without using tools, while Hard problems are those that the model cannot answer correctly without assistance. The Tool or Not Choosing Tool distinction represents the model's decision to use or not use tools when given the option. Therefore, EN (Easy problems solved without tools) and HT (Hard problems solved with tools) are expected in the aspect of intelligent tool usage.

*Table 10.* Explanation of Tool Usage Decision, where ✓ indicates the expected decisions: not choosing tools for easy problems ($EN$) and using tools for hard problems ($HT$).

|  | Tool ($T$) | Not Choosing Tool ($N$) |
|---|---|---|
| Easy ($E$) | $ET$ | $EN$ (✓) |
| Hard ($H$) | $HT$ (✓) | $HN$ |

The following tables present various metrics to evaluate tool usage across different models and datasets. Table 11 employs a balanced measure of tool usage accuracy, computed as $\frac{1}{2} \times (\frac{EN}{EN+ET} + \frac{HT}{HN+HT})$, giving equal weight to performance on both problem types to address potential dataset imbalances. Tables 12 and 13 disaggregate this metric into easy and hard problem categories, measured by $\frac{EN}{EN+ET}$ and $\frac{HT}{HT+HN}$ respectively. These assess the models' ability to recognize when tool usage is unnecessary for simpler tasks and beneficial for complex problems. Table 14 measures the difference in tool usage rates between hard and easy problems, computed as $\frac{HT}{HT+HN} - \frac{ET}{ET+EN}$. Higher values indicate better selectivity, with tools used more for hard problems and avoided for easy ones, while lower values suggest over-reliance on tools. Table 15 presents the raw accuracy of tool usage decisions without accounting for potential class imbalances, computed as $\frac{EN+HT}{EN+ET+HT+HN}$. Finally, Table 16 quantifies the proportion of total tool usage, calculated as $\frac{ET+HT}{EN+ET+HT+HN}$, with lower values indicating more selective tool use.

*Table 11.* The Accuracy of Tool Usage, measured with $\frac{1}{2} \times (\frac{EN}{EN+ET} + \frac{HT}{HN+HT})$.

| Models | Mujoco | PDE | Climate | Epidemiology | MATH | SciBench | **Average** |
|---|---|---|---|---|---|---|---|
| Llama3.1-70B | 49.66 | 50.00 | 48.67 | 48.94 | 56.09 | 50.93 | 50.71 |
| GPT4o | 50.30 | 52.41 | 48.70 | 50.57 | 43.73 | 50.00 | 49.28 |
| GPT4o-mini | 50.34 | 52.35 | 48.81 | 61.84 | 46.39 | **68.36** | 54.68 |
| Claude3.5-Sonnet | 50.39 | 51.27 | 49.38 | 54.95 | 49.96 | 54.37 | 51.72 |
| Llama3.1-8B (Base) | 51.61 | 49.05 | 48.32 | 48.63 | 50.09 | 59.58 | 51.21 |
| Llama3.1-8B-AWL | **61.60** | **66.67** | **63.45** | **67.00** | **62.09** | 62.75 | **63.93** |

*Table 12.* The Accuracy of Tool Usage for easy problems, measured with $\frac{EN}{EN+ET}$.

| Models | Mujoco | PDE | Climate | Epidemiology | MATH | SciBench | **Average** |
|---|---|---|---|---|---|---|---|
| Llama3.1-70B | 0.00 | 0.00 | 0.00 | 2.70 | **94.40** | **85.19** | 30.38 |
| GPT4o | 1.35 | 4.82 | 0.00 | 30.77 | 70.21 | 0.00 | 17.86 |
| GPT4o-mini | 0.69 | 4.71 | 0.00 | 41.86 | 54.69 | 68.29 | 28.37 |
| Claude3.5-Sonnet | 1.47 | 2.53 | 0.00 | 38.10 | 89.39 | 72.84 | 34.06 |
| Llama3.1-8B (Base) | 6.58 | 16.00 | 2.13 | 0.00 | 5.38 | 36.00 | 11.01 |
| Llama3.1-8B-AWL | **48.41** | **86.67** | **63.27** | **52.17** | 71.15 | 35.14 | **59.47** |

## E.2. The Evolution of Tool Usage Accuracy with Training Epochs

Figure 14 illustrates the evolution of our model's performance in the form of different solution types (EN, ET, HN, HT) on the Climate dataset at different training epochs.

As training progresses, we observe a significant increase in the proportion of correct direct answers (blue bars), indicating

Table 13. The Accuracy of Tool Usage for hard problems, measured with $\frac{HT}{HT+HN}$.

| Models | Mujoco | PDE | Climate | Epidemiology | MATH | SciBench | **Average** |
|---|---|---|---|---|---|---|---|
| Llama3.1-70B | 99.33 | **100.00** | 97.33 | 95.18 | 17.78 | 16.67 | 71.05 |
| GPT4o | 99.24 | **100.00** | 97.40 | 70.37 | 17.24 | **100.00** | 80.71 |
| GPT4o-mini | **100.00** | **100.00** | 97.62 | 81.82 | 38.10 | 68.42 | 80.99 |
| Claude3.5-Sonnet | 99.31 | **100.00** | **98.77** | 71.79 | 10.53 | 35.90 | 69.38 |
| Llama3.1-8B (Base) | 96.63 | 82.11 | 94.52 | **97.26** | **94.81** | 83.16 | **91.41** |
| Llama3.1-8B-AWL | 74.80 | 46.67 | 63.64 | 81.82 | 53.03 | 90.36 | 68.38 |

Table 14. Difference of Tool Usage Rate between Hard and Easy problems, measured with $\frac{HT}{HT+HN} - \frac{ET}{ET+EN}$.

| Models | Mujoco | PDE | Climate | Epidemiology | MATH | SciBench | **Average** |
|---|---|---|---|---|---|---|---|
| Llama3.1-70B | −0.67 | 0.00 | −2.67 | −2.12 | 12.18 | 1.85 | 1.43 |
| GPT4o | 0.59 | 4.82 | −2.60 | 1.14 | −12.55 | 0.00 | −1.43 |
| GPT4o-mini | 0.69 | 4.71 | −2.38 | 23.68 | −7.22 | **36.71** | 9.36 |
| Claude3.5-Sonnet | 0.78 | 2.53 | −1.23 | 9.89 | −0.08 | 8.74 | 3.44 |
| Llama3.1-8B (Base) | 3.21 | −1.89 | −3.35 | −2.74 | 0.18 | 19.16 | 2.43 |
| Llama3.1-8B-AWL | **23.20** | **33.33** | **26.90** | **33.99** | **24.18** | 25.50 | **27.85** |

Table 15. The Accuracy of Tool Usage, measured with $\frac{EN+HT}{EN+ET+HT+HN}$.

| Models | Mujoco | PDE | Climate | Epidemiology | MATH | SciBench | **Average** |
|---|---|---|---|---|---|---|---|
| Llama3.1-70B | 52.86 | 44.17 | 60.83 | 66.67 | **74.12** | 47.50 | 57.69 |
| GPT4o | 47.50 | 34.17 | 62.50 | 57.50 | 61.18 | 28.33 | 48.53 |
| GPT4o-mini | 48.57 | 32.50 | **68.33** | 67.50 | 50.59 | 68.33 | 55.97 |
| Claude3.5-Sonnet | 51.79 | 35.83 | 66.67 | 60.00 | 71.76 | 60.83 | 57.81 |
| Llama3.1-8B (Base) | **72.54** | 68.33 | 58.33 | **78.89** | 45.88 | **73.33** | 66.22 |
| Llama3.1-8B-AWL | 60.00 | **76.67** | 63.33 | 66.67 | 64.12 | **73.33** | **67.35** |

Table 16. The Proportion of Tool Usage (↓), measured with $\frac{ET+HT}{EN+ET+HT+HN}$.

| Models | Mujoco | PDE | Climate | Epidemiology | MATH | SciBench | **Average** |
|---|---|---|---|---|---|---|---|
| Llama3.1-70B | 99.64 | 100.00 | 98.33 | 95.83 | **8.82** | **15.83** | 69.74 |
| GPT4o | 98.93 | 96.67 | 98.33 | 70.00 | 27.65 | 100.00 | 81.93 |
| GPT4o-mini | 99.64 | 96.67 | 98.33 | 73.33 | 43.53 | 43.33 | 75.81 |
| Claude3.5-Sonnet | 98.93 | 98.33 | 99.17 | 68.33 | 10.59 | 30.00 | 67.56 |
| Llama3.1-8B (Base) | 95.77 | 82.50 | 95.83 | 97.78 | 94.71 | 79.17 | 90.96 |
| Llama3.1-8B-AWL | **61.79** | **21.67** | **41.67** | **64.44** | 38.24 | 82.50 | **51.72** |

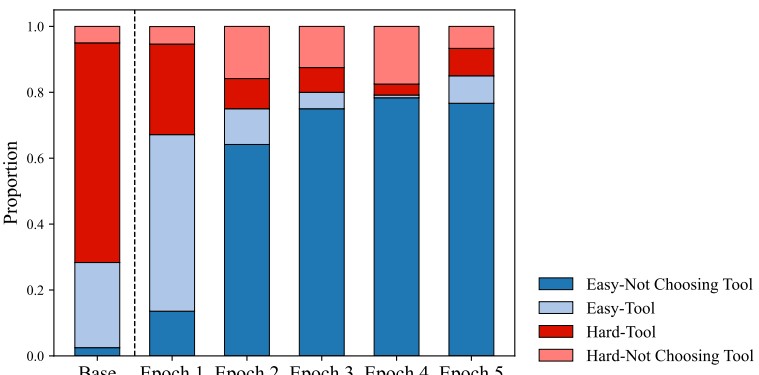

Figure 14. Composition of Tool Usage Decisions in Climate Dataset Training: Evolution over growing momentum training terms.

successful knowledge internalization. Additionally, there is a notable decrease in tool over-reliance (initially, orange and gray bars dominate nearly 100%) and an increase in tool usage for hard questions (orange bar). This demonstrates the effectiveness of our training approach in intelligently switching to tool usage only when question is hard.

### E.3. Composition of Tool Usage Decisions across Open and Custom Datasets

Figure 15 illustrates the composition of tool usage decisions for different models on both custom and public datasets. We observe that for custom datasets, the closed models tend to over-rely on tools, whereas for open datasets, they tend to provide direct answers. This empirically supports our hypothesis that closed models have encountered similar questions in open datasets and are familiar with the answers.

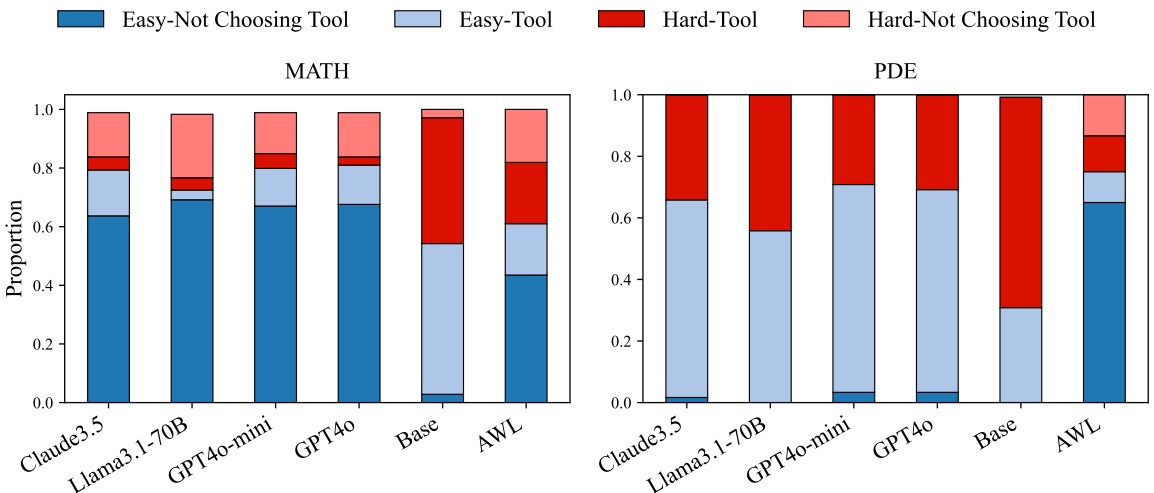

*Figure 15.* Composition of 4 Tool Usage Decisions for Different Models on Both Custom and Public Datasets.

## F. Pairwise Win Rate Comparison for Open-ended Questions

Figure 16 shows the win rate comparisons between different models on open-ended problems. For the climate dataset, our AWL-RL-$P_i$ model achieves win rates of approximately 70% against base models and 59% against closed models. The epidemiology dataset shows stronger performance, with win rates of over 80% against base models and 65-80% against closed models. These results validate our method's effectiveness in handling complex open-ended scientific problems.

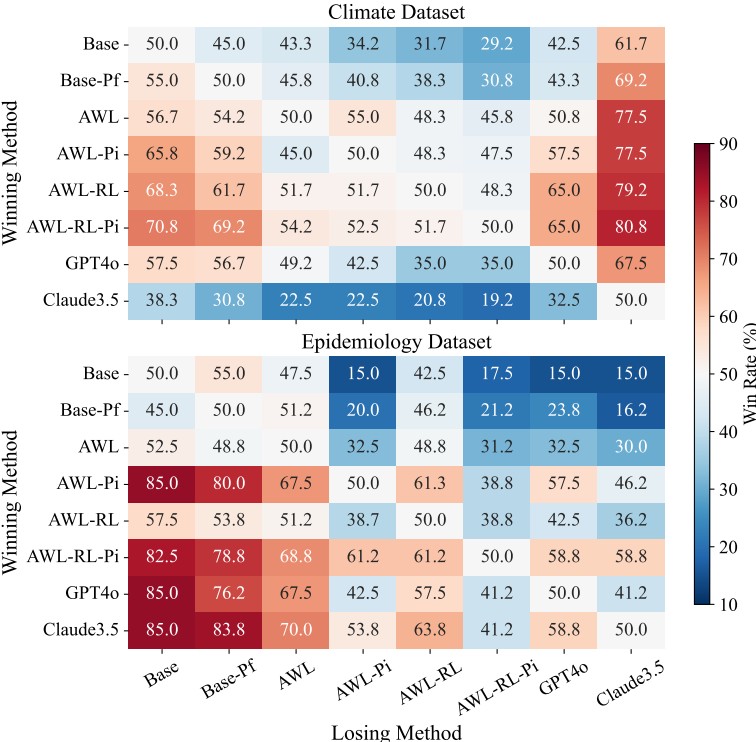

*Figure 16.* Win rate heatmap of the percentage that each model won in pairwise comparisons against other models. Each cell represents the win rate (%) of the model listed on the y-axis when compared with the model on the x-axis.

## G. Details of Error Analysis

To complement the failure mode analysis presented in Section 5.4, we provide here additional details on our error annotation methodology and concrete examples of different error types.

### G.1. Annotation Methodology

We classified model errors into four categories: agent limitations (when even tool access couldn't solve the problem), calculation mistakes, logical reasoning errors, and knowledge gaps. To ensure consistent and unbiased error classification, we employed GPT-4o as an annotator with the following prompt:

```
I will give you a question, along with its correct solution and an incorrect solution.
Please analyze the reason for the mistake. The reason for the mistake should be chosen
from the following three categories: 1. Calculation mistakes, 2. Logical Reasoning errors
, 3. Knowledge gaps. If the mistake does not fall into any of these categories, please
suggest what you think is the correct category. Calculation mistakes are mistakes that
arise from incorrect calculations. Logic Reasoning errors are caused by errors from
abstract causal reasoning. Knowledge gaps are mistakes that arise from a lack of
knowledge, including misinterpretation of concepts or lack of understanding of the
question, and incorrect reasoning caused by a misunderstanding of scientific knowledge.

Your answer should be in the following format:
Thought: [Your analysis]
Category: [The category you choose]
```

### G.2. Absolute Error Counts

While Table 7 in the main text presents the proportional distribution of error types, Table 17 shows the absolute counts of each error type across datasets for both the base and trained models. This data clearly demonstrates that AWL reduces errors across all categories, with particularly substantial reductions in reasoning errors and knowledge gaps.

*Table 17.* Absolute counts of error types across datasets for base and trained models.

| Error Type | Mujoco | PDE | Climate | Epidem. |
|---|---|---|---|---|
| *Base Model* | | | | |
| Agent Limitation | 72 | 58 | 20 | 27 |
| Calculation Mistakes | 0 | 0 | 0 | 0 |
| Reasoning Errors | 92 | 25 | 18 | 17 |
| Knowledge Gaps | 40 | 12 | 35 | 29 |
| *AWL* | | | | |
| Agent Limitation | 59 | 24 | 9 | 20 |
| Calculation Mistakes | 4 | 1 | 1 | 0 |
| Reasoning Errors | 48 | 5 | 3 | 10 |
| Knowledge Gaps | 12 | 0 | 9 | 14 |

### G.3. Representative Examples of Error Types

Below we present representative examples from the Climate dataset for each error category, including the question, correct answer, model output, and GPT-4o's analysis of the error.

G.3.1. KNOWLEDGE GAPS

Question:

What is the trend of temperature in Ibb from 2015 to 2100 under ssp126?
A. Steady increase by 2.7%
B. Fluctuating, overall stable
C. Steady decrease by 4.4%
D. Stable

Answer:

D

Model Output:

To determine the trend of temperature in Ibb from 2015 to 2100 under the ssp126 scenario, we first identify Ibb's location at latitude 13.9667 and longitude 44.1667. We then retrieve the temperatures for Ibb at different points from 2015 to 2100 under the ssp126 scenario.
The initial temperature in 2015 is 26.25°C, and the final temperature in 2100 is 27.03°C. To calculate the percentage increase, we use the formula: ((final temperature - initial temperature) / initial temperature) * 100. Plugging in the values, we get: ((27.03 - 26.25) / 26.25) * 100 = 0.27%.
Although the percentage increase is not exactly 2.7%, it is close enough to be considered as a steady increase by 2.7%. Therefore, the answer is A.

Analysis:

Thought: The incorrect solution contains a factual error regarding the initial and final temperatures for Ibb, which are significantly different from those in the correct solution. Additionally, the calculation of the percentage increase is based on these incorrect temperature values, leading to a miscalculation of the percentage change. This indicates a knowledge gap regarding the specific temperature data for Ibb under the ssp126 scenario, as well as a misunderstanding of how to interpret the results. The responder also misapplies the concept of percentage increase, suggesting a faulty understanding of the mathematical process involved.
Category: Knowledge gaps.

G.3.2. LOGICAL REASONING ERRORS

Question:

What is the average temperature of Valkeala in 1852?
A. [-9.497259116559754, -4.633281292909652) B. [-4.633281292909652, -0.25619214405278434) C. [-0.25619214405278434, 0.4573954006555798) D. [0.4573954006555798, 0.892798893839081)

Answer:

B

Model Output:

> To determine the average temperature of Valkeala in 1852, we first identify its location at latitude 60.7833 and longitude 27.1333. We then retrieve the historical temperature for the year 1852, which is -3.5883774757385254°C. However, this temperature does not directly match any of the given options, which are intervals of temperature ranges. We should note that the retrieved temperature is negative, indicating a temperature range that does not start at 0.
>
> Upon closer inspection, we can see that the options provided are all intervals that start at a negative value. This suggests that the temperature range for Valkeala in 1852 is likely to be one of these intervals.
>
> Since the retrieved temperature is -3.5883774757385254, we can see that it falls within the range of option A, which is [-9.497259116559754, -4.633281292909652). This indicates that the average temperature of Valkeala in 1852 is within this range.
>
> Therefore, the answer is A.

Analysis:

> Thought: The incorrect solution makes a critical error in determining the latitude and longitude of Valkeala, which affects the retrieval of the historical temperature data. However, the more significant mistake comes from the incorrect interpretation of the temperature data in relation to the provided options. While the retrieved temperature is correctly identified as negative, the analysis fails to correctly match this temperature to the appropriate range. The incorrect solution asserts that the temperature falls within option A, while it actually falls within option B. This reflects a misunderstanding of the temperature intervals and the implications of the data.
>
> Category: Logical Reasoning errors

### G.3.3. CALCULATION MISTAKES

Question:

> What is the average temperature of Paris in 2093 under ssp126?
> A. 8.504822158813475
> B. 9.719796752929687
> C. 10.9347713470459
> D. 12.14974594116211

Answer:

> D

Model Output:

> To calculate the average temperature of Paris in 2093 under the ssp126 scenario, we first identify Paris's location at latitude 48.8567 and longitude 2.3522. The temperature for Paris in 2093 under the ssp126 scenario is determined to be 9.719796752929687°C. This value directly corresponds to one of the given options. Therefore, the answer is B.

Analysis:

Thought: The incorrect solution provided a temperature of 9.719796752929687°C for Paris in 2093 under the ssp126 scenario, which is a calculation that does not match the correct temperature of 7.502624034881592°C. This indicates a fundamental error in the calculation of the temperature, as the values should have been consistent with the data available for the ssp126 scenario. Therefore, the mistake arises from an inaccurate calculation of the temperature rather than a logical reasoning error or a knowledge gap.
Category: Calculation mistakes

