# OpenReview forum: "Adapting While Learning: Grounding LLMs for Scientific Problems with Tool Usage Adaptation"
_ICML.cc/2025/Conference — ICML 2025 poster_

### Official Review · Reviewer_qrRN · 2025-03-09

**Overall Recommendation:** 3

**Summary:**

The paper **"Adapting While Learning: Grounding LLMs for Scientific Problems with Tool Usage Adaptation"** introduces **Adapting While Learning (AWL)**, a two-component fine-tuning approach that enables LLMs to intelligently decide when to rely on internal reasoning or external tools for solving scientific problems. The first component, **World Knowledge Learning (WKL)**, trains LLMs to internalize knowledge by learning from tool-generated solutions, while the second, **Tool Usage Adaptation (TUA)**, classifies problems as easy or hard and trains the model to use tools only when necessary. Tested on six scientific benchmarks—including custom datasets in climate science, epidemiology, and mathematics—the method improves answer accuracy by **28.27%** and tool usage accuracy by **13.76%**, even surpassing **GPT-4o** and **Claude-3.5** on specialized tasks. AWL introduces a **more efficient and adaptive problem-solving paradigm**, reducing over-reliance on costly computational tools while enhancing scientific reasoning in LLMs.

**Claims And Evidence:**

The main contribution of this paper are:

(1) we construct 4 datasets that include various scientific domains. The descriptions, and construction method should be clearly stated in the main text. However, I can not find this. These things are stated in supplementary materials.


(2) The second innovation is designed adapting while learning. The two components are: World Knowledge Learning and Tool Usage Adaption. However, I can not understanding why World Knowledge Learning and Tool Usage Adaption can be summarized as adapting while learning. I seems to adapting while inference?

(3) The world knowledge learning and tool usage adaptation are not novel. The first one is very similar to SFT. The tool usage learning method is also largely borrowed from "Learning to Use Tools via Cooperative and Interactive Agents"

Zhengliang Shi, Shen Gao, Xiuyi Chen, Yue Feng, Lingyong Yan, Haibo Shi, Dawei Yin, Pengjie Ren, Suzan Verberne, Zhaochun Ren, Learning to Use Tools via Cooperative and Interactive Agents

**Essential References Not Discussed:**

The paper's key contributions would benefit from discussing related works that have explored similar themes of adaptive tool usage and reasoning in large language models (LLMs).

1. **Program-Aided Language Models (PAL):** Gao et al. introduced PAL, where LLMs generate Python programs to solve complex problems, effectively integrating external computational tools to enhance reasoning capabilities.

2. **Automatic Multi-Step Reasoning and Tool-Use (ART):** Paranjape et al. proposed ART, enabling LLMs to perform multi-step reasoning and decide when to invoke external tools, aligning closely with the adaptive tool usage discussed in the current paper.

3. **TaskMatrix.AI:** Liang et al. developed TaskMatrix.AI, connecting foundation models with millions of APIs, thereby allowing LLMs to interact with external systems and tools to complete various tasks.

4. **Gorilla:** Patil et al. presented Gorilla, an approach where LLMs are connected with massive APIs, facilitating dynamic tool usage based on the task requirements.

**Experimental Designs Or Analyses:**

While providing codes in the supplementary materials, could you provide comprehensive information on the four custom-created datasets, including data collection methods, annotation processes, and validation protocols? Additionally, will these datasets be made publicly available to facilitate reproducibility and further research?



How do you define and measure 'tool usage accuracy'? Does this metric assess the model's ability to decide appropriately between internal reasoning and tool usage, or does it evaluate the correctness of tool-assisted answers?​


What specific criteria or thresholds do you use to classify problems as 'easy' or 'hard' within the TUA component? How consistent are these classifications across different domains and datasets?

**Methods And Evaluation Criteria:**

Tool Usage Accuracy: The metric for tool usage accuracy is not clearly defined. It is unclear whether this metric assesses the model's ability to choose correctly between internal reasoning and tool usage or if it measures the correctness of tool-assisted answers. A precise definition of this metric is crucial for interpreting the results accurately.


The overall method is reasonable. However, these methods seem to be not novel, as discussed in Claims And Evidence.

Also:
The TUA component is designed to enable the model to decide when to use external tools based on problem difficulty. However, the criteria for classifying problems as 'easy' or 'hard' are not thoroughly explained. A clear understanding of this classification is essential to assess the validity of the tool usage decisions and the overall effectiveness of the TUA mechanism.

**Other Comments Or Suggestions:**

NO

**Other Strengths And Weaknesses:**

The most weakness of this paper is its novelty and its writing.

**Questions For Authors:**

At this time, I do not have additional issues to raise beyond those already discussed.

**Relation To Broader Scientific Literature:**

The paper's key contributions align with existing research on enhancing large language models (LLMs) through adaptive tool usage and internal knowledge integration. The proposed **Adapting While Learning (AWL)** framework, comprising **World Knowledge Learning (WKL)** and **Tool Usage Adaptation (TUA)**, mirrors efforts in **Retrieval-Augmented Generation (RAG)**, where LLMs access external information to improve accuracy and reduce hallucinations. Additionally, the concept of LLMs autonomously determining when to utilize external tools parallels advancements in creating AI agents capable of reasoning and acting, such as the **ReAct** pattern, which integrates LLMs as planners that "think out loud" before executing actions. By constructing new scientific datasets across various domains, the paper contributes to the broader endeavor of grounding LLMs in specialized knowledge, enhancing their problem-solving capabilities in complex scientific contexts.

**Theoretical Claims:**

NO theoretical claims.

---

> ### Author Rebuttal · Authors · 2025-03-31
>
> Thank you for your thoughtful review. We address your concerns below:
>
> ---
>
> ##  Information on the custom-created datasets
>
> > "...The descriptions, and construction method should be clearly stated in the main text. However, I can not find this...."
>
> > "Could you provide comprehensive information on the four custom-created datasets, including data collection methods, annotation processes, and validation protocols?"
>
> In our current paper, key information about four custom-created datasets is already presented in Section 4.1 and Figure 2 in the main text. The construction methods are detailed in Lines 220-230 and illustrated in Figure 2. More detailed information is presented in Appendix A. We will enhance clarity in our revised paper and guide readers to the corresponding sections.
>
> > “Additionally, will these datasets be made publicly available to facilitate reproducibility and further research?”
>
> Yes, the datasets and code will be made public upon publication.
>
> ## Clarification of “Adapting While Learning”
>
> > "... However, I can not understanding why World Knowledge Learning and Tool Usage Adaption can be summarized as adapting while learning. I seems to adapting while inference?"
>
> "Adapting While Learning" refers to adaptation occurring during the knowledge-learning process. The "learning" refers to the World Knowledge Learning phase where the model learns scientific knowledge through tool interactions.
>
> Both adapting (tool usage adaptation) and learning (world knowledge learning) are built into the loss term of the fine-tuning process (Eq 6), not at inference time. If one were to create "adapt while inference," they would need additional procedures (e.g., Monte Carlo Tree Search) during inference, which is not the case in our framework.
>
> ## Novelty of our work and relationship with previous literature
>
> > "The world knowledge learning and tool usage adaptation are not novel. The first one is very similar to SFT."
>
> There appears to be a misunderstanding. Our innovation is NOT about SFT technology, but in adapting it to enable a new LLM-based learning paradigm for scientific problem-solving: learning knowledge from scientific simulators and learning to decide when to seek help from tools depending on problem complexity.  Similar to how human scientists approach a problem, our training paradigm enables LLMs to use scientific simulation tools adaptively. We demonstrated strong empirical performance in several scientific domains including climate science, epidemiology, etc.
>
> > “The tool usage learning method is also largely borrowed from 'Learning to Use Tools via Cooperative and Interactive Agents'”
>
> Our work is fundamentally different from theirs. The work of Shi et al. (2024) primarily focuses on constructing datasets to enhance tool usage ability, i.e., they only focus on tool calling, which can lead to over-reliance on tool calls (see our results in Table 2). This potential over-reliance is actually a motivation for our development of this work.
>
> ## Definition of Tool Usage Accuracy
>
> > "The metric for tool usage accuracy is not clearly defined."
>
> > "How do you define and measure 'tool usage accuracy'? ..."
>
> Tool usage accuracy is clearly defined in Section 4.3 (Lines 321 left col. - 303 right col.) in the main text and we provide potential 5 additional tool usage metrics in Appendix E.  This metric assesses the model's ability to choose correctly between internal reasoning and tool usage. We can simplify our writing to better guide readers in our revised version.
>
> ## Easy/Hard Problem Classification
>
> > "The criteria for classifying problems as 'easy' or 'hard' are not thoroughly explained."
>
> > "What specific criteria or thresholds do you use to classify problems as 'easy' or 'hard' within the TUA component? How consistent are these classifications across different domains and datasets?"
>
> We describe this classification in Lines 186-193 and provide additional details for open-ended questions in Lines 248-254. Questions are classified as "easy" or "hard" based on the model's direct-answer accuracy—if a model answers correctly without tools, it's "easy"; otherwise, "hard." This classification is dynamic across datasets and models, but the rule for dividing is consistent. We will simplify our writing for a more concise and clear information delivery in the revised paper.
>
> ## References Not Discussed
>
> Thank you so much for these suggested references. We will discuss them in the "LLM Tool Usage" paragraph in Section 2 of the revised paper. We kindly note that Gorilla is already included (Line 20, Line 100, right col).
>
> ---
>
> Thank you again for your time and effort. We hope our response addresses your concerns. If the issues have been resolved, we’d appreciate your consideration in the evaluation. Please feel free to share any additional feedback.

---

> > ### Comment · Reviewer_qrRN · 2025-04-04
> >
> > I have read the rebuttal carefully. The rebuttal address my concerns.

---

### Official Review · Reviewer_n1ey · 2025-03-12

**Overall Recommendation:** 4

**Summary:**

This paper presents Adapting While Learning i.e AWL, a novel fine-tuning method to improve the performance of LLMs on scientific problems by adaptively using external tools based on question complexity. It tackles the issue of either hallucinations obtained thorugh fine tuning or excessive reliance on tools by training models in two stages: World Knowledge Learning for internalizing solutions provided by tools, and Tool Usage Adaptation for adaptive tool usage decisions based on problem difficulty. The authors evaluate the approach using six benchmarks spanning mathematics, physics, epidemiology, and climate science, achieving significant accuracy improvements and outperforming state-of-the-art models such as GPT-4o and Claude-3.5. Four new scientific datasets are developed to further test the effectiveness of the proposed method. The results demonstrate the AWL model substantially improves both answer accuracy (28.27%) and tool usage accuracy (13.76%), surpassing state-of-the-art models on complex, custom datasets.

**Claims And Evidence:**

**Claim:** The authors claim that the AWL improves accuracy and optimizes tool usage decisions adaptively

**Evidence:** Empirical evaluations show significant accuracy improvements (up to 28.27% higher answer accuracy and 13.76% improved tool accuracy)

**Claim**: AWL significantly outperforms both baseline and state-of-the-art LLMs on custom and challenging scientific benchmarks


**Evidence**: Comparison against state-of-the-art models (GPT-4o and Claude-3.5) on both public (MATH, SciBench) and four custom-created scientific datasets supports the claim that the proposed method outperforms existing models, especially on specialized, novel datasets.

**Essential References Not Discussed:**

No

**Experimental Designs Or Analyses:**

* The paper conducts extensive experiments, comparing multiple baselines (including different configurations of Llama3.1, GPT-4o, Claude-3.5) on diverse scientific datasets.
* Ablation studies clearly demonstrate that both components (WKL and TUA) are necessary and effective.
* The study on robustness to noise is critical and thoughtfully executed, further validating the method's resilience against real-world uncertainties and data variability.
* The design for open-ended questions, involving preference optimization (DPO), effectively addresses the complexity inherent to problems lacking definitive single solutions by explicitly evaluating solution proposals based on predefined domain metrics.

**Methods And Evaluation Criteria:**

* Supervised Fine-tuning and Preference Optimization methods are used to teach LLMs the internal scientific knowledge generated from tools.
* Tool Usage Adaptation (TUA) classifies the problems into "easy" and "hard" based on the LLM’s performance without tools, adapting the training approach accordingly.
* The answer accuracy is measured based on correct responses to both multiple-choice and numerical-answer problems.
* The tool usage accuracy evaluates whether the model correctly decides when to use or skip tools based on the complexity classification (easy/hard) of the problem, ensuring efficiency and reliability

**Other Comments Or Suggestions:**

N/A

**Other Strengths And Weaknesses:**

## Strengths

The paper is well written and addresses a very important problem regarding LLMs use for scientific research

The Novel framework considered for this papers is useful and makes sense, getting some of the inspiration from the way humans handle the problems and then select the tools.

The models selection, the methodology along with the experiments conducted are detailed in depth

## Weaknesses

May be they could have considered more variations in loss functions and also showed some result with some experiments why is the proposed loss function better than the existing one and what are the other possible ones.

Also increase the partition to not just easy/hard but to more sections .

Also it would be interesting to see prompt tuning/prompt injection as comparison where the initial query is considered and based on the tool usage prompt is updated/injected according to the tool usage. This would remove the fine tuning part and make it little easier.

**Questions For Authors:**

N/A

**Relation To Broader Scientific Literature:**

The contributions are indeed very useful, this research encourages further exploration of enabling LLMs to adapt to tool usage specially in the AI for Science field which would help improve the performance of the LLMs and also they can better assist humans in research.

**Theoretical Claims:**

I did check the theoretical claim, there is one claim about the loss function in the paper on line 194 and it makes sense to use ensemble method to avoid the performance degradation.

---

> ### Author Rebuttal · Authors · 2025-03-31
>
> Thank you for your invaluable feedback. We address your suggestions below:
>
> ---
>
> ## Variations in loss functions
>
> > "May be they could have considered more variations in loss functions and also showed some result with some experiments why is the proposed loss function better than the existing one and what are the other possible ones."
>
> We thank you for this suggestion. Our work focuses on defining a new paradigm for LLMs to learn to solve scientific questions adaptively; as long as that target can be fulfilled, the loss term definition (also, similarly, the specific finetuning frame) is orthogonal to our work. We kindly find this ablation is not strongly related to our work.
>
> ## Increasing Problem Partitions
>
> > "Also increase the partition to not just easy/hard but to more sections."
>
> This is a very insightful observation and a great suggestion for future work. In this work, we are the first of our kind to encourage LLMs to choose smartly when to use tools, hence the setting is simplistic and binary.
>
> But surely, in realistic scientific or engineering cases, we can have different granularity of tools for different requirements of a given problem.
>
> As noted in Lines 420–424 (right col.), we have already acknowledged this limitation and discussed the potential of extending the framework to support more fine-grained partitions in future work. We appreciate the reviewer highlighting this direction and will further discuss it.
>
> ## Functionality of Prompt Tuning
>
> > "Also it would be interesting to see prompt tuning/prompt injection as a comparison where the initial query is considered and based on the tool usage prompt is updated/injected according to the tool usage."
>
> We appreciate this suggestion. In our current paper, Appendix B already shows prompts instructing models (without finetuning) to decide tool usage based on question difficulty.
>
> To further investigate the functionality of prompt tuning (PT) on these tasks, we added additional experiments using few-shot prompting with direct-answer and tool-usage examples. We show its performance compared with the base model and our method on answer accuracy and tool usage accuracy.
>
> **Answer Accuracy**
>
> ||Mujoco|PDE|Climate|Epidemiology|MATH|SciBench|Avg.|
> |-|-|-|-|-|-|-|-|
> |Base|57.14|59.17|76.67|58.89|55.89|29.17|56.16|
> |PT|61.43|59.17|76.67|58.89|49.41|26.07|55.27|
> |Ours|**64.17**|**78.33**|**83.33**|**74.44**|**62.35**|**34.17**|**66.13**|
>
> **Tool Usage Accuracy**
>
> ||Mujoco|PDE|Climate|Epidemiology|MATH|SciBench|Avg.|
> |-|-|-|-|-|-|-|-|
> |Base|51.50|50.00|50.75|50.86|50.09|60.22|52.24|
> |PT|54.08|50.00|50.96|48.63|53.19|55.09|51.99|
> |Ours|**61.80**|**66.67**|**75.50**|**66.61**|**62.09**|**62.75**|**65.90**|
>
> The experimental results indicate that few-shot prompting does not consistently improve the model's performance and may even degrade it in most cases. We will include the above results as a baseline in both Table 1 and Table 2 in our revised paper.
>
> ---
>
> Thank you again for your time and effort. Our paper benefits from your suggestions. We hope our response addresses your concerns. We welcome any further comments and are happy to address them.

---

> > ### Comment · Reviewer_n1ey · 2025-04-04
> >
> > Thank you for the feedback, I would like to keep my score.

---

### Official Review · Reviewer_tT3B · 2025-03-13

**Overall Recommendation:** 3

**Summary:**

This paper proposes a new fine-tuning method called “Adapting While Learning (AWL)” that addresses the challenges of using large language models (LLMs) to solve scientific problems. LLM is effective for simple scientific problems, but can hallucinate on complex problems, and while integration with external tools is a solution, models fine-tuned specifically for tool use tend to be unnecessarily dependent on the tools even for simple problems. AWL is a takes a two-stage approach inspired by the way human experts evaluate the complexity of a problem before choosing a solution. In the first stage, “World Knowledge Learning (WKL)”, the LLM internalizes scientific knowledge using answers generated by the tool. In the second stage, “Tool Usage Adaptation (TUA)”, the model classifies problems as “easy” or “difficult” based on the accuracy of its direct answers, and trains the model to respond directly to easy problems and use the tool for difficult problems. In experiments using six datasets from fields such as climate science, epidemiology, and mathematics, the model with AWL achieved an average of 28.27% improvement in response accuracy and 13.76% improvement in tool usage accuracy compared to the baseline model, and in four custom-made datasets, it outperformed state-of-the-art models such as GPT-4o and Claude-3.5.

## Post-Rebuttal Update

After reviewing the authors' rebuttal, I find that they have adequately addressed the concerns raised in my review. The authors provided additional information on:

1. Validation on larger models: They conducted experiments with Qwen2.5-14B-Instruct, showing consistent performance improvements with their method on larger models.

1. Hyperparameter sensitivity analysis: They provided a sensitivity analysis on the number of samples (k=1,3,5) used to assess problem difficulty, demonstrating the stability of their approach across different parameter settings.

1. Computational cost reduction analysis: They added analysis on tool usage frequency in open-ended problem-solving, showing significant reductions in unnecessary tool use.

1. Error analysis: They introduced a new error analysis section categorizing error types, which shows that after training, the proportion of reasoning errors and knowledge gaps decreased, while errors stemming from "agent limitations" (problems unsolvable even with tools) increased.

These additional analyses further support the robustness and practicality of the proposed method. The authors have also effectively addressed concerns raised by other reviewers. I maintain my original assessment and recommendation for this paper.

**Claims And Evidence:**

The main claim of the paper is that the proposed AWL approach improves LLM's scientific problem-solving ability and enables adaptive switching of tool use according to the difficulty of the problem. This claim is well supported by the following evidence:

In terms of response accuracy, the AWL-trained model outperforms the base model on all datasets, showing an average accuracy improvement of 28.27%. In particular, the authors' custom dataset outperforms even state-of-the-art models such as GPT-4o and Claude-3.5.

In terms of tool use accuracy, the AWL model achieves an average accuracy of 65.90%, compared to other models (which achieve around 50% tool use accuracy). The difficulty analysis of the MATH dataset demonstrates the AWL model's ability to increase tool use according to the difficulty of the problem.

Ablation experiments show that both WKL and TUA components are necessary for optimal performance, confirming that a single component alone is not sufficient. Experiments on robustness to noise show that the AWL model maintains response accuracy even at high noise levels.

Experiments on extending the model to open-ended questions also show that the model combining AWL and DPO performs well in generating constrained responses.

This evidence is clear and convincing, supporting the claims made in the paper.

**Essential References Not Discussed:**

The paper focuses on the specific issue of using adaptive tools to solve scientific problems, and the relevant key literature is appropriately cited and discussed. However, the context could be strengthened by adding further references to the following points

1. The cognitive science literature on metacognition and self-assessment: The paper cites the work of Kruger & Dunning, but a reference to recent research on LLM self-assessment (e.g. Kadavath et al., “Language models (mostly) know what they know”) could help to provide a deeper understanding of the challenges involved in models assessing their own abilities.

2. Unsupervised and self-supervised problem difficulty evaluation: By referring to recent research on methods for LLM to autonomously evaluate the difficulty of problems (e.g. Sun et al., “Self-Evaluation Guided Beam Search for Reasoning”), The authors can consider alternative approaches to the TUA component.

3. Causal effects of tool use: By referring to research on causal understanding when LLM uses tools (e.g., Yao et al., “React: Synergizing reasoning and acting in language models”), it may be possible to consider more effective tool use strategies.

Although the absence of these references does not cause a critical problem in understanding the main contribution of the paper, it would help to place it in a wider research context.

**Experimental Designs Or Analyses:**

The experimental design and analysis are sound overall. The reviewer verified the following points:

1. **Diversity of data sets**: The six data sets used cover a variety of scientific domains, including climate science, epidemiology, and PDE, and are suitable for evaluating the generality of the method.

2. **Choice of baseline**: Using Llama-3.1-8B as the base model and comparing it with state-of-the-art models such as GPT-4o, GPT-4o-mini, Claude-3.5-Sonnet, and Llama-3.1-70B is appropriate.

3. **Evaluation metrics**: In addition to the two main metrics of response accuracy and tool usage accuracy, metrics have also been introduced to evaluate the appropriateness of tool usage for both simple and difficult questions, allowing for a multifaceted evaluation.

4. **Ablation experiments**: Ablation experiments have been conducted to evaluate the individual effects of WKL and TUA, demonstrating the necessity of both components.

5. **Robustness to noise**: The robustness of the proposed method is evaluated through experiments in which the noise level of the training data is varied.

The only concern is that the validation is limited to models other than Llama and large-scale models (10B+). It would also have been better if the sensitivity analysis of hyperparameters (especially the threshold that determines the difficulty of the problem) had been carried out in more detail.

**Methods And Evaluation Criteria:**

The proposed method and evaluation criteria are appropriate for the application of scientific problem solving.

The proposed method, AWL, attempts to capture the essence of actual scientific reasoning, which is the adaptive use of tools according to the complexity of the problem. It models the natural behavioral pattern of human scientists, who directly solve simple problems and rely on computational tools for complex problems, and this is reasonable in the context of scientific problem solving.

Two main metrics are used as evaluation criteria: accuracy of answers and accuracy of tool use, which are suitable for evaluating the effectiveness of the method from multiple perspectives. In particular, the accuracy of tool use is defined as “the ability to use the tool for difficult problems and not use the tool for easy problems”, which directly corresponds to the purpose of the proposed method.

The evaluation dataset also uses custom datasets that cover a wide range of scientific domains, including climate science, epidemiology, and PDEs, in addition to general benchmarks such as MATH and SciBench, making it suitable for evaluating the generality of the method. In addition, by using datasets that include problems of different difficulty levels, the adaptive tool-use ability can be appropriately evaluated.

Furthermore, to evaluate the scalability of the method to open-ended problems, the method combined with DPO was also evaluated, which corresponds to realistic scenarios of scientific problem solving.

**Other Comments Or Suggestions:**

Suggestions for further improving the quality of the paper:

1. Adding a sensitivity analysis of hyperparameters (especially the problem difficulty threshold) would further demonstrate the stability and generality of the proposed method.

2. Adding a quantitative analysis of computational cost reduction would further demonstrate the practical benefits of the proposed method.

3. Adding an analysis of knowledge transfer between different datasets and domains would provide further insight into the model's generalization ability.

4. By adding validation on larger models (10B+) and a variety of models, you can demonstrate the scalability of the proposed method.

5. By adding an error analysis section and analyzing the patterns of cases where the proposed method fails, you can identify areas for future improvement.

6. By adding an evaluation of the quality of answers to open-ended questions by human experts, you can more strongly demonstrate the practical value of the proposed method.

7. By providing detailed descriptions of application cases and usage scenarios in actual scientific research environments, the practical significance of the proposed method can be more clearly demonstrated.

**Other Strengths And Weaknesses:**

**Strengths:**

1. **Practical solutions**: The authors propose practical approaches that correspond to practical scenarios of scientific problem solving, from simple to complex problems.

2. **Original ideas**: Our approach is inspired by human cognitive processes, and provides a new perspective on adaptive tool use according to the complexity of the problem.

3. **Comprehensive experiments**: The effectiveness of the proposed methods is verified from multiple perspectives, including evaluation across diverse scientific domains, ablation experiments, and noise robustness verification.

4. **Dataset contribution**: New scientific problem-solving datasets are constructed and released for the research community.

5. **Extension to open-ended problems**: Extensions are proposed to handle open-ended scientific problems beyond canned answers.

**Weaknesses:**

1. **Lack of verification of generality and scalability**: The effectiveness of the proposed method has only been verified to a limited extent on LLM other than Llama and on larger-scale models (10B+).

2. **Knowledge transfer between domains**: A more convincing result would have been obtained if there had been a detailed analysis of knowledge and ability transfer between different scientific domains.

3. **Real-world scientific research applications**: What are the specific examples (use cases) of the target problem in this research? The paper provides conceptual explanations, but does not provide detailed descriptions of specific use case scenarios. There is limited detailed examination of the applicability of the proposed method in actual scientific research environments.

5. **Evaluation by human experts**: There is a lack of qualitative evaluation by human experts, particularly regarding the quality of answers to open-ended questions.

**Questions For Authors:**

I look forward to responses to the concerns I have raised above, but I have no further questions.

**Relation To Broader Scientific Literature:**

The contributions of this paper are related to three research areas: LLM alignment, LLM training for scientific problem solving, and LLM tool use.

As LLM alignment techniques, the paper adopts SFT (supervised fine-tuning) and DPO (direct preference optimization), which are based on existing studies such as Rafailov et al. (2024) and Ouyang et al. (2022).

While existing studies on LLM training for scientific problem solving rely on expert annotation or distillation from strong models (Thulke et al., 2024; Zhang et al., 2024b), our study proposes an automatic knowledge acquisition approach using tools.

In terms of LLM tool use, while existing research such as Toolformer by Schick et al. (2023) proposes methods for teaching LLM specific tool use patterns, this study is new in that it focuses on the ability to adaptively determine tool use according to the complexity of the problem.

In particular, Yu et al. (2024) pointed out the lack of adaptability in LLM's tool-use decisions, and this study directly addresses this issue. It is also interesting that it takes inspiration from human cognitive science (Payne et al., 1993; Kruger & Dunning, 1999) and incorporates the insight that humans evaluate the complexity of a problem before choosing a solution.

**Theoretical Claims:**

This paper focuses mainly on experimental methods and results, and does not include any major theoretical arguments, including rigorous theoretical proofs. However, it does provide a conceptual explanation of the design principles of AWL and a logical basis for why both WKL and TUA are necessary. These explanations are intuitive and consistent with the experimental results.

---

> ### Author Rebuttal · Authors · 2025-03-31
>
> Thank you for your detailed review and constructive suggestions. We address your concerns below.
>
> ---
>
> ## Validation on Larger Models
>
> > ...limited to models other than Llama and large-scale models(10B+)
>
> We additionally included Qwen2.5-14B-Instruct and trained it with our method. We conducted experiments on 2 custom and 2 public datasets. We present models’ answer accuracy under $P\_n$ and $P\_i$, Tool Usage Accuracy (TUA), and Tool Use Ratio (TUR), which corresponds to Table1, 2, 13 in the paper respectively.
>
> |Method|PDE|Mujoco|MATH|SciBench|
> |-|-|-|-|-|
> |Base-$P\_n$|61.67|54.28|74.12|17.50|
> |Ours-$P\_n$|78.33|60.00|81.18|56.67|
> |Base-$P\_i$|69.17|44.28|79.41|46.67|
> |Ours-$P\_i$|80.00|62.85|82.35|65.83|
> |Base-TUA|48.91|50.00|48.45|48.84|
> |Ours-TUA|63.58|54.16|54.69|58.54|
> |Base-TUR (↓)|99.17|100.00|95.88|93.33|
> |Ours-TUR (↓)|13.33|27.14|1.76|44.17|
>
> Results show consistent gains in answer and tool usage accuracy, with reduced tool use. It shows our method also benefits larger models.
>
> ## Problem Difficulty Hyperparameters
>
> > ...a sensitivity analysis of hyperparameters (especially the problem difficulty threshold)...
>
> We conducted a sensitivity analysis on the number of samples (k = 1, 3, 5) used to assess LLMs' answer accuracy and partition questions by difficulty. We trained different models with these thresholds on MATH and SciBench datasets, with the results presented below.
>
> Answer Accuracy (pass@k)
>
> |k|MATH(Base)|SciBench(Base)|MATH(Ours)|SciBench(Ours)|
> |-|-|-|-|-|
> |1|54.71|17.50|62.09|30.83|
> |3|65.88|30.00|72.35|54.16|
> |5|74.11|37.50|75.88|55.83|
>
> Tool Usage Accuracy
>
> |k|MATH(Base)|SciBench(Base)|MATH(Ours)|SciBench(Ours)|
> |-|-|-|-|-|
> |1|50.09|60.22|62.09|62.75|
> |3|57.73|52.38|64.37|58.74|
> |5|62.16|52.22|65.36|58.27|
>
> Results show that our method remains stable across different hyperparameters about the problem difficulty threshold.
>
> ## Computational Cost Reduction
>
> > ...a quantitative analysis of computational cost reduction...
>
> Currently, Tables 4 and 13 have already presented the tool use ratio of different models and showed that our method can reduce unnecessary tool usage thus reducing computational cost.
>
> Additionally, we conducted an analysis on open-ended problem-solving which allows multiple tool usage. The numbers in the table represent the average times of tool usage across all questions. It further demonstrates that our method brings computational cost reduction.
>
> ||Base|Ours|
> |-|-|-|
> |Climate|7.21|2.70|
> |Epidemiology |2.80|0.42|
>
> ## Error Analysis
>
> > ...error analysis and case patterns...
>
> We add a new section on error analysis. We categorize errors into: 1. Problems unsolvable even with tool usage (agent limitation); 2. Problems solvable with tools but answered incorrectly. The latter category is further divided into: a. Calculation mistakes, b. Reasoning errors, and c. Knowledge gaps.
>
> Following standard practices in benchmark papers, we provide GPT-4o with the errors and their corresponding correct solutions and ask it to annotate their error types. The error type distributions are read below.
>
> Base model
>
> ||Mujoco|PDE|Climate|Epidemiology|MATH|SciBench|
> |-|-|-|-|-|-|-|
> |Agent Limitation|35.29|35.83|27.40|36.99|45.45|29.47|
> |Calculation Mistakes|0.00|0.00|0.00|0.00|2.60|7.37|
> |Reasoning Errors|45.10|55.83|24.66|23.29|46.75|56.84|
> |Knowledge Gaps|19.61|8.33|47.95|39.73|5.19|6.31|
>
> Our trained model
>
> ||Mujoco|PDE|Climate|Epidemiology|MATH|SciBench|
> |-|-|-|-|-|-|-|
> |Agent Limitation|47.97|80.00|40.91|45.45|49.23|32.53|
> |Calculation Mistakes|3.25|3.33|4.55|0.00|4.62|7.23|
> |Reasoning Errors|39.02|16.67|13.64|22.73|41.54|54.22|
> |Knowledge Gaps|9.76|0.00|40.91|31.82|4.62|6.02|
>
> After training, the proportion of Reasoning Errors and Knowledge Gaps decreased and the proportion of Agent Limitation increased. This suggests that: 1. Our method enables the model to learn scientific reasoning and domain-specific knowledge; 2. The remaining errors after training are mainly concentrated on questions that cannot be correctly solved even with tool usage. This analysis points out the key challenges that future work should address.
>
> ## Potential Knowledge Transfer
>
> Cross-domain generalization isn't the primary focus of this work. We have discussed it in our future work section (Line 416-420, right col.). We will further highlight it.
>
> ## Application Descriptions
>
> We have provided examples of application cases and usage scenarios in Figure 2 and Appendix A (Pages 12-18). We’ll clarify application relevance in revisions.
>
> ## Human Evaluation for Open-ended Questions
>
> For these questions, our evaluation is based on the simulation result given the LLMs' proposals, which is intrinsically stable. Manual reviews are out of the scope of this work, but we will include some discussion.
>
> ---
>
> Thank you again for your time and effort. We hope our response has addressed your concerns. If the issues have been resolved, we’d appreciate it if you could reflect it in your evaluation. We welcome any further comments.

---

### Official Review · Reviewer_qGtx · 2025-03-13

**Overall Recommendation:** 3

**Summary:**

The paper introduces a two-component fine-tuning approach that first trains a model on direct reasoning (WKL) and then selectively incorporates tool usage based on the assessed complexity of the scientific problem. The method is empirically validated across six scientific benchmark datasets, demonstrating improved efficiency and accuracy, balancing direct reasoning and tool utilization based on task difficulty.

**Claims And Evidence:**

The claim of novelty and methodological depth is not strongly supported, as the method primarily leverages straightforward dataset-splitting strategies and existing fine-tuning technologies. Additionally, claims about performance advantages are inconsistently supported, with high performance only clearly demonstrated on self-created datasets where details like dataset split methodology and comparison settings (different prompts) are unclear, limiting a fair and thorough evaluation against existing methods.

**Essential References Not Discussed:**

No

**Experimental Designs Or Analyses:**

The approach of splitting datasets into "easy" and "hard" problems based solely on the WKL-trained model's accuracy may lead to inconsistent splits for different models, potentially affecting comparability and fairness. The method achieves higher performance primarily on self-constructed datasets but underperforms on public benchmarks. The authors did not clearly specify the configuration (P_n, P_i, P_f) used for baseline models in their comparisons.

**Methods And Evaluation Criteria:**

The proposed methods and evaluation criteria generally make sense. However, the method's reliance on a fixed, accuracy-based split of easy and hard problems may limit generalizability and robustness, as complexity could vary significantly depending on the model and context.

**Other Comments Or Suggestions:**

I increased my rating one nudge after the discussion phase.

**Other Strengths And Weaknesses:**

Please address the concerns raised in the questions.

**Questions For Authors:**

1. Regarding tool usage adaptation, the methodology involves splitting the dataset into easy and hard instances and training the model with different traces (with/without tool use). While intuitive, this approach lacks theoretical guarantees. The model primarily learns patterns from the dataset, but its reliability diminishes when the dataset changes or when the question format shifts (e.g., from multiple-choice to direct QA). A more robust strategy is needed to address this issue.

2. The performance across datasets and models in Table 1 appears inconsistent. The proposed method achieves the highest scores on self-made datasets (Mujoco, PDE, Climate, Epidemiology), whereas GPT-4o performs best on public datasets (MATH, SciBench). One possible explanation is that GPT-4o is unfamiliar with these tasks and custom tools, while the proposed model has been specifically trained on them. Additionally, it is unclear which setting (P_n, P_i, P_f) is used for other models.

3. Concerns also arise regarding tool usage accuracy. First, the settings (P_n, P_i, P_f) used for different models are not explicitly stated. Second, the easy/hard split should vary across models, but it is unclear whether the authors use a fixed or dynamic split. Lastly, not all scientific tools are resource-intensive (e.g., Python scripts); in such cases, the relevance of the proposed metric remains questionable.

**Relation To Broader Scientific Literature:**

The proposed method aligns closely with existing concepts of conditional computation, adaptive inference, and selective prediction. Although practically useful, the approach mainly applies established techniques without significantly extending them.

**Theoretical Claims:**

There is no proof for theoretical claims.

---

> ### Author Rebuttal · Authors · 2025-03-31
>
> Thank you for your thoughtful review. We address your concerns below:
>
> ---
>
> ## Novelty and Methodological Depth
>
> > "The claim of novelty and methodological depth ... existing fine-tuning technologies."
>
> We are not claiming novelty in fine-tuning technologies. Rather, our innovation primarily lies in defining a novel learning paradigm for scientific problem-solving with LLMs, i.e., learning knowledge from scientific simulators and switching to tools intelligently based on problem complexity, which fits human expert behavior in scientific domains. We are the first to propose this paradigm, and our empirical experiments demonstrate that our paradigm can strike a great balance between answer accuracy and tool calling cost.
>
> ## Performance Consistency on Public Datasets
>
> > "...claims about performance advantages are inconsistently supported...", "...in Table 1...GPT-4o performs best on public datasets."
>
> The performance difference on public datasets is largely due to model size, as the strongest baselines such as GPT-4o are significantly larger than our base model (8B). To support this, we additionally include 2 larger open-source models Qwen2.5-{14B/32B}-Instruct as base models for our method and conduct experiments on MATH and SciBench, with results presented below, where TUA means Tool Usage Accuracy:
>
> |Model|MATH(14B)|SciBench(14B)|MATH(32B)|SciBench(32B)|
> |-|-|-|-|-|
> |Base-$P\_n$|74.12|17.50|81.77|60.83|
> |Base-$P\_i$|79.41|46.67|84.71|65.83|
> |Base-TUA|48.45|48.84|47.49|48.55|
> |Ours-$P\_n$|81.18|56.67|84.71|62.50|
> |Ours-$P\_i$|82.35|65.83|**85.89**|**69.17**|
> |Ours-TUA|54.69|**58.54**|**55.42**|56.69|
>
> It can be observed that our framework applied to a 32B model can outperform GPT-4o on MATH and achieve comparable performance on SciBench. We will append these results in our revised paper.
>
> ## Dataset Splitting
>
> > "...the easy/hard split should vary across models, but it is unclear whether the authors use a fixed or dynamic split..."
>
> Yes, our easy/hard splitting is relative to each model's problem-solving ability, i.e., dynamic. We described it in Lines 186-193 and provided additional details for open-ended questions in Lines 248-254.
>
> > "...splitting datasets...may lead to inconsistent splits for different models... lacks theoretical guarantees."
>
> The intuition behind this design decision is that different models, by construction, have different levels of problem-solving ability; hence, a fixed split is actually counter-intuitive. If one were to define a fixed split, it would inherently introduce biases using the reference for such a split, regardless of whether that reference is a human expert or an LLM.
>
> As for theoretical guarantees, may you clarify what kinds of theoretical guarantees you are referring to? In Section 3.3, we have already stated that our methods ensure models learn the correct decision patterns through supervised training objectives. Our experiments empirically verify this, which demonstrates that the model successfully internalizes these patterns and applies them consistently. Extra theoretical analysis is rare and trivial in NLP application works.
>
> ## Robustness to Format and Dataset Shifts
>
> > "...its reliability diminishes when the dataset changes or when the question format shifts (e.g., from multiple-choice to direct QA)."
>
> Our datasets already include numerical problems (Lines 231-241) and open-ended questions (Lines 241-247, Figure 3b), which are both direct QAs. Experimental results on these questions are consistent with those on multi-choice questions. It demonstrates our method's robustness to problem format shifts.
>
> ## Experiment Settings
>
> > "...which setting ($P\_n$, $P\_i$, $P\_f$) is used for other models."
>
> In Table1, Llama3.1-70B-Instruct, GPT4o, GPT4o-mini, Claude3.5-Sonnet are vanilla models, i.e., they were evaluated without tool assistance ($P\_n$). We will clarify this in the final version.
>
> ## Relevance of Tool Usage Metric for Lightweight Tools
>
> > "Not all scientific tools are resource-intensive (e.g., Python scripts)..."
>
> We agree that not all scientific tools are resource-intensive, but many of them are, especially in realistic scientific and engineering applications.
>
> These applications, such as climate modeling, and epidemic prediction, would require much more significant computing time than LLM inference (e.g., days for simulation vs. seconds for LLM inference), due to inherent complexity (such as high spatial resolution). Our work targets those use cases and serves as a starting point to apply LLM automation.
>
> Apart from those cases, we also incorporated commonly used benchmarks such as MATH and SciBench, and Python is the most commonly used tool for them. The strong performance of our method shows that our pipeline is applicable to any tools, regardless of their cost.
>
> ---
>
> Thank you again for your time and effort. We hope our response resolves your concerns and would be grateful for your consideration in the evaluation. We welcome any further comments.

---

> > ### Comment · Reviewer_qGtx · 2025-04-05
> >
> > Thanks authors for their clarifications.
> > I maintain that:
> > -The proposed learning paradigm primarily focuses on defining the model’s input and output, which leans more towards an engineering contribution.
> >
> > -Comparing the proposed method directly with the base model may be inappropriate. A more reasonable comparison would be against a standard SFT model under the same tool usage conditions.
> >
> > -Format and Dataset Shifts refer to scenarios where the model is trained on one format or dataset but evaluated on another. For example, the model might be trained on multiple-choice questions but tested on direct question answering using the same dataset. This setting evaluates the model’s robustness in making appropriate tool usage decisions.
> >
> > -The comparison in Table 1 may be unfair if the baseline models are not allowed to use tools, while the proposed method is. A significant portion of the improvement may stem from tool usage rather than the learning paradigm itself.

---

> > > ### Author Response · Authors · 2025-04-09
> > >
> > > We’re glad to hear that your previous questions on performance consistency, dataset splits, theoretical guarantees, and tool usage metrics have been addressed. We now respond to your new points in detail.
> > >
> > > ---
> > >
> > > ## Contribution Type and Significance
> > >
> > > > "... focuses on defining the model’s input and output, which leans more towards an engineering contribution."
> > >
> > > While we don't claim novelty in model training techniques, our key contribution is introducing a novel learning paradigm for scientific problem-solving with LLMs, which goes beyond engineering. This paradigm enables LLMs to learn scientific knowledge through simulator interaction and make intelligent tool selection decisions based on problem complexity.
> > >
> > > We believe research contributions can take various forms including: model architecture design, theoretical analysis, and new paradigms for existing problems. Our work falls inside the last category. We believe it would be inappropriate to discount a work just because it lacks contributions to a certain category, as few papers contribute to all of them.
> > >
> > > We also kindly note that our work’s practicality for scientific problem-solving has been supported by Reviewers tT3B, n1ey and qrRN who acknowledge our work's novelty.
> > >
> > > ## Baseline Comparison
> > >
> > > > "Comparing the proposed method directly with the base model may be inappropriate. A more reasonable comparison would be against a standard SFT model under the same tool usage conditions."
> > >
> > > We appreciate this suggestion and would like to highlight that **we have already performed such an ablation study in Section 5.2** of the paper. Specifically, we evaluated:
> > >
> > > 1. Base model without fine-tuning
> > > 2. SFT model trained with only scientific knowledge distillation
> > > 3. SFT model trained with only intelligent tool usage
> > > 4. Our full method combining both components
> > >
> > > Setting 3 corresponds to the suggested SFT baseline with tool usage. The method underperforms ours in both answer accuracy and tool usage accuracy. The results of this ablation study demonstrates the necessity and contribution of both World Knowledge Learning (WKL) and Tool Usage Adaptation (TUA) components in our proposed method for optimal performance.
> > >
> > > ## Inference-time Dataset Shifts
> > >
> > > > "Format and Dataset Shifts refer to scenarios where the model is trained on one format or dataset but evaluated on another..."
> > >
> > > Thank you for this insight. We acknowledge the importance of studying cross-format generalization. However, it would require significantly more diverse training data than feasible in a single study. Our current work focuses on establishing a new paradigm for scientific problem-solving with LLMs. Cross-format generalization is out of this single work’s capacity but can naturally be a future work direction.
> > >
> > > ## Fairness of Comparisons
> > >
> > > > "The comparison in Table 1 may be unfair ... the improvement may stem from tool usage rather than the learning paradigm itself."
> > >
> > > We want to clarify that Table 1 presents fair and controlled comparisons under different tool usage settings:
> > >
> > > 1. Under $P_n$ (no tools) setting, our model outperforms both the base model and closed-source models. In this comparison, **the improvement clearly stems from the learning paradigm rather than tool usage**, since no tools are allowed for any of the models being compared.
> > > 2. Under $P_i$/$P_f$ (with tools) settings, compared with the base model, our method (1) uses tools more intelligently and selectively, (2) achieves higher answer accuracy with fewer tool calls. This supports that our approach improves both tool usage intelligence and the model's problem-solving ability under tool-using setting.
> > >
> > > We did not include tool-augmented results for closed-source since the goal of our work is not to fine-tune models to use tools more accurately, thus this comparison is unnecessary. Furthermore, even dedicated open-source tool-using LLMs [1, 2] underperform SOTA closed-source models due to model size limitations.
> > >
> > > However, to further address the question, we conducted additional experiments where we compared our model with gpt-4o under tool-augmented settings ($P\_i$) on four datasets. The results show that our trained model performs competitively with GPT-4o, reaffirming the effectiveness of our approach.
> > >
> > > |  | PDE | Mujoco | MATH | SciBench |
> > > | - | - | - |- | - |
> > > | GPT-4o | 82.50 | 75.00 | 79.32 | 50.83 |
> > > | Llama-3.1-8B-Instruct (Base) | 59.17 | 57.14 | 55.89 | 29.17 |
> > > | Llama-3.1-8B-Instruct (Ours) | 78.33 | 64.17 | 62.35 | 34.17 |
> > > | Qwen2.5-14B-Instruct (Base) | 69.17 | 44.28 | 79.41 | 46.67 |
> > > | Qwen2.5-14B-Instruct (Ours) | 80.00 | 62.85 | 82.35 | 65.83 |
> > >
> > > We will clarify these comparisons in our revision.
> > >
> > > ---
> > >
> > > Thank you again for your continued engagement. We hope the above responses address your questions.
> > >
> > > [1] Qin et al., ToolLLM: Facilitating Large Language Models to Master 16000+ Real-world APIs, ICLR 2024 (Spotlight)
> > >
> > > [2] Zeng et al., AgentTuning: Enabling Generalized Agent Abilities for LLMs, ACL 2024

---

### Decision · Program_Chairs · 2025-05-01

**Decision:**

Accept (poster)

**Comment:**

Summary

This paper introduces Adapting While Learning (AWL), a two-component fine-tuning method to improve how Large Language Models (LLMs) solve scientific problems. The first component, World Knowledge Learning (WKL), helps LLMs internalize scientific knowledge by learning from tool-generated solutions. The second component, Tool Usage Adaptation (TUA), trains models to solve simple problems directly while using tools only for complex ones, mimicking human expert behavior. The method was evaluated on six scientific benchmark datasets across climate science, epidemiology, and mathematics, showing substantial improvements in both answer accuracy (28.27%) and tool usage accuracy (13.76%) compared to baseline models, even outperforming state-of-the-art models like GPT-4o and Claude-3.5 on custom datasets.

Reasons to accept

- The paper addresses an important practical problem in AI for science - balancing direct reasoning with tool usage for scientific problem-solving (tT3B, n1ey).
- The approach is inspired by human cognitive processes, providing a novel perspective on adaptive tool use based on problem complexity (tT3B).
- The empirical validation is comprehensive, with experiments across diverse scientific domains, ablation studies, and robustness verification (tT3B, n1ey).
- The method demonstrates significant performance improvements with 28.27% higher answer accuracy and 13.76% better tool usage accuracy (qGtx, tT3B, n1ey).
- The work contributes new scientific problem-solving datasets covering multiple domains that will be made publicly available (tT3B, qrRN).
- Additional experiments showed that the method scales well to larger models (14B and 32B parameters), with consistent performance gains (tT3B).

Recommended revisions

1. The criteria for classifying problems as "easy" or "hard" could be more thoroughly explained (qGtx, qrRN).
2. The cross-domain knowledge transfer capabilities could be explored better
3. The easy/hard split may not be fine-grained enough for real-world scientific problems that might require different granularity of tools (n1ey) -- The authors acknowledged this limitation and discussed it as a potential future work direction.